# Remodeling the cellular stress response for enhanced genetic code expansion in mammalian cells

Mikhail E. Sushkin [1,2,5], Christine Koehler[1,3,5] & Edward A. Lemke [1,4] ✉

Genetic code expansion (GCE) reprograms the translational machinery to site-specifically incorporate noncanonical amino acids (ncAAs) into a selected protein. The efficiency of GCE in mammalian cells might be compromised by cellular stress responses, among which, the protein kinase R(PKR)-dependent eIF2α phosphorylation pathway can reduce translation rates. Here we test several strategies to engineer the eIF2α pathway and boost the rate of translation and show that such interventions increase GCE efficiency in mammalian cells. In particular, addition of the N-terminal PKR fragment (1–174) provides a substantial enhancement in cytoplasmic GCE and also in GCE realized by OTOs (orthogonally translating designer organelles), which built on the principle of 2D phase separation to enable mRNA-selective ncAA incorporation. Our study demonstrates an approach for improving the efficiency of GCE and provides a means by which the power of designer organelles can be further optimized to tune protein translation.

Genetic code expansion (GCE) is a powerful tool that enables site-specific incorporation of noncanonical amino acids (ncAAs) into a protein of interest (POI). In GCE, the amber stop codon (UAG) is most commonly commandeered to extend the capacity of the genetic code and to direct the position of the newly introduced ncAA with codon precision. In most cases, a pair consisting of an orthogonal aminoacyl-tRNA synthetase (RS) and its cognate suppressor tRNA is used to reprogram a stop codon. The synthetase aminoacylates the suppressor tRNA, which recognizes the stop codon introduced into the mRNA sequence, with the desired ncAA. The charged suppressor tRNAs are used by the host translational machinery to provide ncAA incorporation at the sites of introduced stop codons in the POI (Fig. 1a). A key criterion for this to work is that the introduced suppressor tRNA/RS pair is orthogonal to the canonical tRNA/RS pairs in the host machinery, i.e., there is no cross reactivity.

However, GCE, in which the tRNA/RS pair essentially swims in the cytoplasm, is not mRNA-selective, and, in principle, naturally occurring stop codons in mRNAs coding for host proteins can be suppressed.

Approximately 20% of stop codons in mammalian cells are amber codons, a concern that can lead to background incorporation and nontargeted and largely uncharacterized impacts upon the cell. The use of orthogonally translating organelles (OTOs) provides a means to address this issue in mammalian cells[1–3]. Here, inspired by the concept of phase separation, an organelle is designed to specifically perform protein engineering. The fundamental idea of such a system lies in creating a spatially distinct place in the cell with a unique biochemical composition, in which the suppressor tRNA and the mRNA of the POI are highly concentrated, so that protein translation preferentially executes amber suppression only in or near the organelle. In these systems, GCE is also mRNA-selective. OTOs work particularly well if grafted to surfaces[2], with the resulting thin-film environment that imposes very high concentration of the suppressor tRNA, which is potentially facilitated by 2D condensation principles.

More than 200 different bespoke ncAAs with new functionalities can be encoded into a protein, and this has been found to be useful for, for example, labeling a protein with fluorescent dyes by means of click

[1]Biocenter, Johannes Gutenberg University Mainz, Hanns-Dieter-Hüsch-Weg 17, 55128 Mainz, Germany. [2]International PhD Programme of the Institute of Molecular Biology, Hanns-Dieter-Hüsch-Weg 17, 55128 Mainz, Germany. [3]VERAXA Biotech GmbH, Carl-Friedrich-Gauß-Ring 5, 69124 Heidelberg, Germany. [4]Institute of Molecular Biology gGmbH, Ackermannweg 4, 55128 Mainz, Germany. [5]These authors contributed equally: Mikhail E. Sushkin, Christine Koehler. ✉e-mail: edlemke@uni-mainz.de

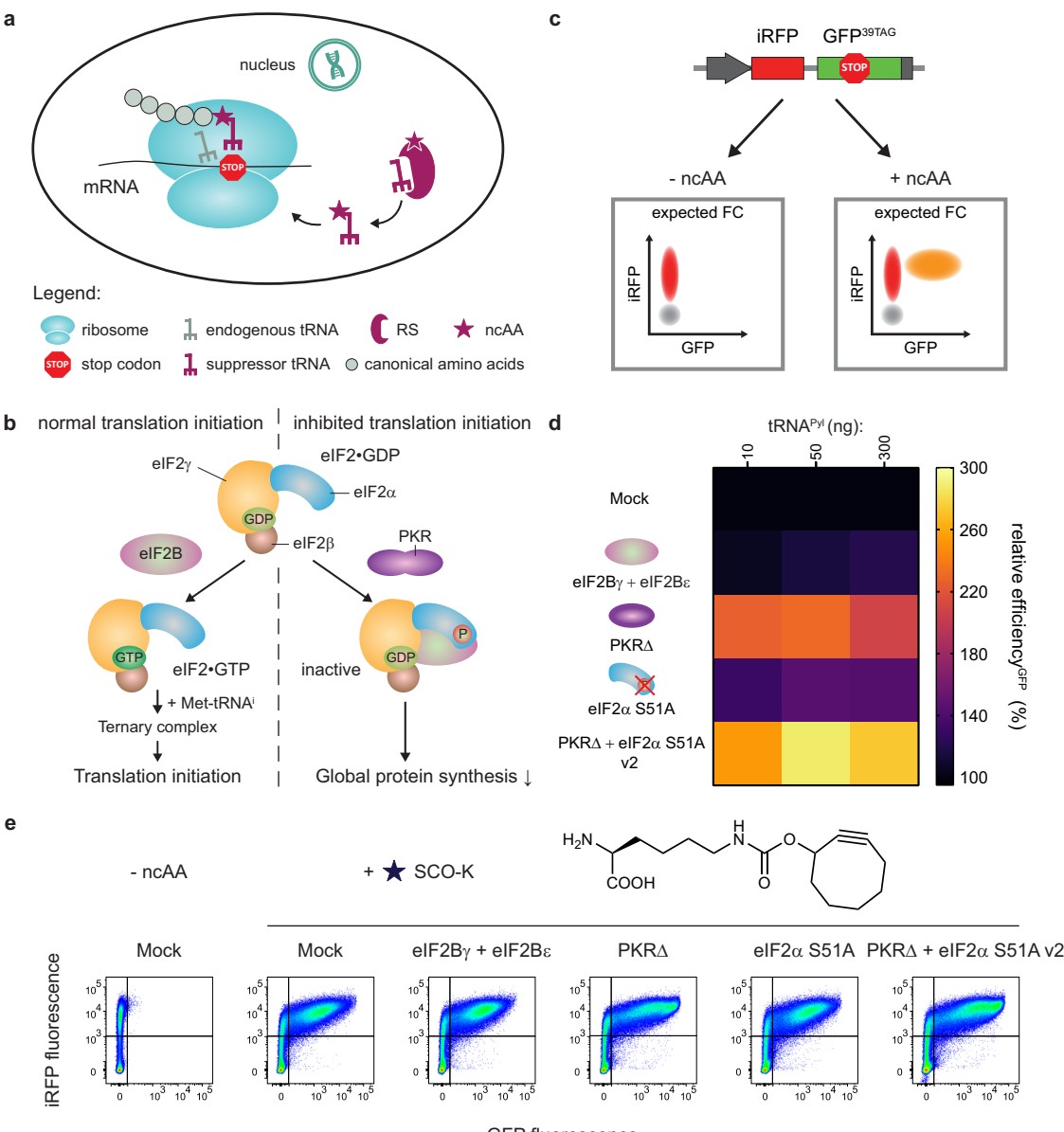

**Fig. 1 | Reprogramming the PKR-dependent eIF2α phosphorylation pathway enables GCE enhancement in mammalian cells. a** General scheme of GCE. RS designates aminoacyl-tRNA synthetase, ncAA - noncanonical amino acid. **b** Schematic representation of the PKR-dependent eIF2α phosphorylation pathway. **c** Schematic representation of the iRFP-GFP^39TAG reporter. The gray population represents untransfected cells; gray arrow – CMV promoter; gray box – polyA signal, FC denotes flow cytometry. **d** Heat map illustrating the effect of adding various stress remodelers on GCE efficiency in units of relative efficiency^GFP (%).

Percentage relative efficiency^GFP is calculated as the median GFP signal for each particular case divided by the median GFP signal for control samples with the addition of mock plasmid. Median GFP signals were obtained after FC analysis of corresponding samples. The heat map shows the mean value for the relative efficiencies of three independent experiments. Source data are provided as a Source Data file. **e** FC analysis of the iRFP-GFP^39TAG reporter in the absence and presence of the tested stress remodelers. Concatenated data from three independent experiments (50 ng tRNA^Pyl plasmid) are shown.

chemistry[4], antibody engineering[5], or manipulating post-translational modifications[6] (see refs. 7,8 for reviews and references therein). All these applications rely on high GCE efficiency. As GCE competes with the natural machinery to terminate translation if a stop codon is encountered, it generally suffers from lower yields compared with the expression of wild-type proteins. Typically, cases where 50% of wild type (WT) protein yield is achieved for a ncAA-modified protein are often considered near the optimum, while the average is much lower. To overcome the almost intrinsic lower yields, several approaches have been proposed for enhancing GCE in prokaryotes and eukaryotes. In addition to the more conventional strategies of simply using strong promoters, increasing copy numbers of genes, and so forth, the most common approaches include: (1) evolution of RS and tRNA;

(2) discovery and development of new orthogonal RS/tRNA pairs; (3) genome engineering; and (4) modifications of the translational machinery, such as ribosomes, elongation factors, the nonsense-mediated decay pathway, and release factors (as thoroughly reviewed in refs. 9–14).

Here, we focus on another crucial process that can severely affect protein synthesis in mammalian cells—the cellular stress response—and present another strategy for improving GCE efficiency in mammalian cells by engineering the stress-induced pathway to boost translation.

GCE efficiency in mammalian cells might be lowered by cellular responses to stress conditions; in particular, a high concentration of RNA can activate the protein kinase R (PKR)-dependent eIF2α

phosphorylation pathway, which is a part of the wide cascade of reactions united under the term integrated stress response (ISR). In a normal state, the eIF2•GDP complex is activated through the exchange of GDP for GTP performed by the eIF2B complex (Fig. 1b, left). Then, the eIF2•GTP complex interacts with the initiator Met-tRNA$^i$ to form a ternary complex, which further transforms into a preinitiation complex to promote the initiation of translation. Under stress conditions, for example, a viral infection, PKR senses high RNA levels, undergoes homodimerization and autophosphorylation and finally, phosphorylates eIF2α at Ser51 (sequence number 52 according to UniProt P05198, Fig. 1b, right). Phosphorylated eIF2α acts not as a substrate but as an inhibitor of eIF2B, assembling a stable inactive complex. This blocking ultimately leads to no activation of the eIF2 complex, inhibition of translation initiation, and a global decrease in the rate of translation. Notably, PKR can also be activated by many different factors (e.g., heat shock proteins, growth factors, and heparin) and by a number of processes including ER stress, oxidative stress, and bacterial infection. It also downstream regulates inflammation, cell metabolism, and apoptosis, and is one of the central hubs in the cellular stress response (for reviews, see refs. 15,16).

In this study, we test three strategies for harnessing the PKR-dependent eIF2α phosphorylation pathway to improve GCE efficiency in mammalian cells: (1) addition of catalytic subunits of the eIF2B complex (subunits γ and ε); (2) introduction of the N-terminal PKR fragment (1–174; PKRΔ); and (3) implementation of mutated eIF2α S51A. Strategy 1 implies an increase in the guanine nucleotide exchange factor (GEF) activity of the eIF2B complex and aims to increase the abundance of the eIF2•GTP complexes. It has been shown that expressed in mammalian cells and purified subunits eIF2Bγ and eIF2Bε (which are considered to be responsible for the catalytic activity of the eIF2B complex) retain 20% of GEF activity in an in vitro assay when used as the binary complex eIF2Bγε[17]. Strategy 2 relies on the dominant negative function of PKRΔ, which contains two double-stranded RNA (dsRNA) binding domains but lacks the catalytic domain of a wild-type PKR. Several mechanisms can contribute to the negative PKRΔ effect: (a) PKRΔ can bind RNAs and sequester them from full-length PKR, preventing PKR activation; (b) the interaction of PKRΔ with full-length PKR can drive the formation of heterodimers with lower kinase activity; and (c) PKRΔ competes with full-length PKR for ribosome binding, thus decreasing full-length PKR access to eIF2α coupled with ribosomal subunits. The presence of PKRΔ causes a reduction in PKR autophosphorylation and eIF2α phosphorylation, ultimately increasing reporter protein expression in transfection assays[18–20]. Strategy 3 aims to erase the phosphorylation site in eIF2α via the mutation S51A and to outcompete endogenous eIF2α in potential phosphorylation reactions to lower the concentration of phosphorylated eIF2α. Such an approach has been demonstrated to increase the protein yield in transfected mammalian cells[21].

In this paper, we show that effectors (namely stress remodelers) from all three strategies help to increase GCE efficiency in mammalian cells with the best system allowing up to 2.8-fold GCE enhancement for single amber suppression. We further test the most efficient stress remodelers from the strategies for improving multiple ncAA incorporation into the POI and gain up to 3.2-fold GCE enhancement. Next, we apply stress remodeling strategies to increase the yield of ncAA-modified antibody trastuzumab, used for generating antibody-drug conjugates (ADCs)[5,22–24], and production of which is costly and thus yield sensitive. Addition of PKRΔ stress remodeler enables 1.7-fold improvement in production of ncAA-modified trastuzumab illustrating its potential for use in biotechnological applications. Finally, we examine the best performing strategies for enhancing mRNA-specific GCE facilitated by film-like OTOs and achieve up to 1.8-fold increase in GCE efficiency for single amber suppression and up to 3.2-fold GCE improvement for multiple ncAA incorporation. This study demonstrates how tuning the stress response can be used to enhance GCE in

mammalian cells, and how this can also be applied to modernize the functionality of designer membraneless OTOs.

## Results

### Stress remodeling strategies enable GCE enhancement

To evaluate the proposed strategies, we tested GCE enhancement in mammalian cells using the orthogonal PylRS/tRNA$^{Pyl}$ pair from *Methanosarcina mazei*, which is one of the most commonly used pairs for GCE, and cyclooctyne-lysine (SCO-K) as a ncAA. We refer to this system as the cytoplasmic PylRS system to distinguish it from the film-like OTOs studied below. HEK293T cells were transiently transfected with distinct plasmids containing NES-PylRS$^{AF}$, tRNA$^{Pyl}$, and the fluorescent reporter fusion iRFP-GFP$^{39TAG}$ [4,25], and GCE efficiency levels in the absence or presence of various potential stress remodelers were compared using flow cytometry (FC). The reporter is designed such that iRFP is expressed regardless of the GCE performance. However, GFP fluorescence can be detected only after incorporation of the ncAA at position 39 of GFP (Fig. 1c). For this reporter, GCE efficiency is defined as the median GFP signal for each particular FC dataset divided by the median GFP signal for samples with the addition of mock plasmid; we call this calculated parameter the relative efficiency$^{GFP}$ (%). For example, a relative efficiency$^{GFP}$ (%) of 200 means that we achieved a twofold increase in GCE efficiency. As GCE is highly dependent on suppressor tRNA levels, typically, in this paper, we investigated various stress remodelers at different amounts of tRNA$^{Pyl}$ plasmid: 10, 50, and 300 ng (Fig. 1d, e, Supplementary Figs. 1–4), unless otherwise noted. As shown in Fig. 1d, all three remodeling strategies afforded an increase in GCE efficiency relative to conditions without additional stress remodelers (i.e., transfection with mock plasmid). For strategy 1, only the combination of the two catalytic subunits eIF2Bγ and eIF2Bε provided a low additional GCE signal, but individually none of them did (Fig. 1d, e, Supplementary Figs. 1, 2). Adding PKRΔ according to strategy 2 showed up to a 2.3-fold enhancement of the GFP signal (Fig. 1d, e, Supplementary Figs. 1, 3). The mutated version, PKRΔ K64E, which disrupts dsRNA-binding activity[26], demonstrated a lower level of GCE enhancement, consistent with previous findings regarding the dependence of the translation boost on RNA binding[18]. Introduction of eIF2α S51A following strategy 3 led to a remarkable (1.4-fold) increase in GCE efficiency, but to a smaller extent compared with PKRΔ (2.3-fold, Fig. 1d, e, Supplementary Figs. 1, 4, Supplementary Note 1). Incorporation of ncAA into iRFP-GFP$^{39TAG}$ was confirmed for the expression in the absence as well as in the presence of the best-performing stress remodelers from all three strategies using LC-MS/MS analysis (Supplementary Figs. 5–8).

Next, we checked if a combination of the two most efficient strategies (strategy 2 and 3, respectively) could lead to an even higher boost in GCE efficiency. Three further different designs of constructs were examined: (i) PKRΔ and eIF2αS51A were cloned individually into the bidirectional promoter vector (PKRΔ + eIF2α S51A v1); (ii) fusion construct PKRΔ-P2A-eIF2α S51A was assembled (PKRΔ + eIF2α S51A v2), where P2A is a peptide responsible for ribosome skipping and thus for separation of stress remodelers; and (iii) fusion construct with inverted order of stress remodelers chained via P2A sequence was designed – eIF2α S51A-P2A-PKRΔ (namely PKRΔ + eIF2α S51A v3). Gratifyingly, application of PKRΔ + eIF2α S51A v2 provided a higher 2.8-fold enhancement of the GFP signal (Fig. 1d, e, Supplementary Figs. 1, 9). Tailoring the cellular stress pathway might potentially lead to disturbance of the translational machinery and cause GCE-independent readthrough of stop codons in mRNAs encoding a POI. Such inaccuracy in translation might give rise to the incorporation of endogenous amino acids instead of selected ncAAs at the sites of introduced stop codons. To assess the potential undesired readthrough of the amber stop codon in the mRNA encoding a POI, we tested the effect of stress remodelers in the absence of ncAA. We observed only a small low-efficient

GFP population in the absence of ncAA, indicating that untargeted readthrough is not a major concern in our experiments (Supplementary Fig. 10). Intensity of this population was also suppressor tRNA-dependent, meaning that a decrease in tRNA$^{Pyl}$ level led to a corresponding drop in GFP signal. These experiments confirm accurate ncAA incorporation after addition of various stress remodelers.

Changes in the iRFP signal were also analyzed for all tested stress remodelers to assess iRFP expression levels (Supplementary Fig. 1). For this we quantified relative efficiency$^{iRFP}$ (%), defined as the median iRFP signal for each particular FC dataset divided by the median iRFP signal for samples with the addition of mock plasmid. A small increase (up to 1.3-fold) in the iRFP signal was observed compared with the up to 2.8-fold increase in the GFP amber suppression signal. However, we found that an increase in the iRFP signal always correlates with an increase in the GFP signal (Supplementary Fig. 1). We then investigated how GCE enhancement is correlated with general translation boost after addition of stress remodelers in another reporter systems. First, the iRFP-GFP$^{39TAG}$ reporter was titrated to 10, 50, and 300 ng of transfected reporter plasmid, and experiments with the addition of NES-PylRS$^{AF}$, titrated tRNA levels, and PKRΔ were performed. Usage of 50 ng of the reporter led to similar fold changes in the iRFP and GFP signals, while at 10 ng of the reporter fold change for iRFP was even higher for lower tRNA levels to compare with the fold increase in the GFP signal (Supplementary Figs. 11 and 12). Second, we applied another fluorescent reporter, termed GFP-P2A-T2A-mCherry$^{189TAG}$, where T2A is a known ribosome skipping peptide (scheme of the reporter is illustrated in Supplementary Fig. 13a). Introduction of GFP-P2A-T2A-mCherry$^{189TAG}$ reporter leads to a constant GFP and GCE-dependent mCherry expression from the same mRNA. For experiments with such a reporter we calculated relative efficiency$^{mCherry}$ (%) and relative efficiency$^{GFP}$ (%), dividing the median mCherry or GFP signal of a sample by the median mCherry or GFP signal respectively for samples with the addition of mock plasmid. Addition of stress remodelers led to a similar or even lower fold increase in the mCherry signal to compare with fold increase in the GFP signal (e.g., 1.6-fold and 1.9-fold in case of PKRΔ + eIF2α S51A v2 addition for mCherry and GFP respectively, Supplementary Figs. 13–15). Taken together, for all tested reporters we could observe an increase in both GCE efficiency and general translation, and therefore all studies with titrated iRFP-GFP$^{39TAG}$ and GFP-P2A-T2A-mCherry$^{189TAG}$ validate the applicability of our strategies to enhance GCE.

We also examined how tuning the cellular stress response with developed strategies influences the ISR status. Two key parameters of the ISR status were evaluated—level of phosphorylated eIF2α and global protein synthesis rate[27–30]. Indeed, remodeling the cellular stress response enabled to achieve lower level of eIF2α phosphorylation and higher overall protein synthesis rate (Supplementary Figs. 16, 17, 49, 50, see details in Supplementary Note 2), consistent with the proposed impact of tuning the PKR-dependent eIF2α phosphorylation pathway on the ISR and with measured data.

Stress remodelers were also investigated for the ability to enhance GCE for incorporation of multiple ncAAs into the one POI. Either double-amber reporter iRFP-GFP$^{39TAG,149TAG}$ (2xTAG reporter) or triple-amber reporter iRFP-GFP$^{39TAG,149TAG,182TAG}$ (3xTAG reporter, Fig. 2a) was transfected together with NES-PylRS$^{AF}$ and 50 ng or 300 ng tRNA$^{Pyl}$ and tested in the presence or absence of best stress remodelers from different strategies. Introduction of the combined stress remodeler PKRΔ + eIF2α S51A v2 yielded the highest increase of 3.2-fold in the GFP signal for the double-amber reporter and 2.5-fold for the triple-amber reporter (Fig. 2b, c, Supplementary Figs. 18–22). Together, these results illustrate the possibility for GCE enhancement for the installation of one or multiple ncAAs in diverse systems by engineering the PKR-dependent eIF2α phosphorylation pathway.

## PKRΔ increases production of ncAA-modified antibody

One of the highest potential applications of GCE is the production of ncAA-modified antibodies that are necessary for the synthesis of antibody-drug conjugates (ADCs), used for targeted treatment of various cancer types (reviewed in Ref. [31,32]). Trastuzumab (antibody binding to the HER2 receptor) was chosen due to its wide-spread usage for ADC synthesis for improved therapy against breast cancer[33,34]. For testing if the addition of stress remodelers helps to increase the yield of ncAA-modified trastuzumab, two vectors were assembled: one containing a secretion peptide fused to the light chain of trastuzumab, namely trastuzumab LC, and second containing the secretion peptide fused to the heavy chain of trastuzumab, where the amber stop codon at position 121 was introduced (namely trastuzumab HC$^{121TAG}$, designs of constructs are presented in Fig. 3a). Trastuzumab constructs, combined NES-PylRS$^{AF}$ + tRNA$^{Pyl}$ plasmid, and mock or stress remodeler plasmids were transfected into HEK293T cells, and after incubation with SCO-K antibody concentration in supernatants was measured using an enzyme-linked immunosorbent assay (ELISA). PKRΔ demonstrated the highest, up to 1.7-fold increase in SCO-K-modified antibody concentration (Fig. 3b). No full-length antibody production was detected when SCO-K was not added to the samples. We then aimed to assess the fidelity of ncAA incorporation into trastuzumab in the presence of PKRΔ in mammalian cells. Purified trastuzumab samples were analyzed to determine the intact mass of trastuzumab HC$^{121SCO-K}$. Only one dominant peak at the expected mass value was observed in the MS spectrum, validating precise SCO-K incorporation into the trastuzumab HC (Supplementary Fig. 23). Together, these results demonstrate that PKRΔ introduction can boost the expression yield of ncAA-modified POI and show the potential of PKRΔ usage in GCE-involved biotechnological systems.

## PKRΔ improves GCE efficiency in film-like OTOs

Thus far, we designed OTOs as film-like, "blob"-like, or fiber-like compartments, which are organized in a cell through a combination of cellular spatial targeting and phase separation[1–3,35,36] (Fig. 4a illustrates examples of membrane-associated OT film-like organelles). Rationally designed fusion proteins form a molecular framework for the OTO and contain the following:

a. organelle assemblers, including distinct spatial targeting signals and an intrinsically disordered protein, to direct organelles and facilitate concentration enhancement due to potential phase separation phenomenon;

b. RNA-binding proteins for selective recruitment into the organelle of mRNA modified with corresponding RNA loops [e.g., ms2 bacteriophage coat protein (MCP) as an RNA-binding protein that recognizes ms2 loops];

c. *M. mazei* PylRS to recruit and locally aminoacylate the suppressor tRNA$^{Pyl}$ within the organelle (Fig. 4a).

In addition, OTOs remain easily accessible for the host translational machinery. The mRNA of POIs is preferentially translated with an expanded genetic code, as those are actively recruited to the OTOs by engineering loops into the untranslated region of the mRNA, which bind the corresponding RNA-binding protein in the OTOs (such as ms2 loops which bind tightly to MCP)[1–3]. A unique property of OTOs is the co-localization of selected mRNA translation and suppressor tRNA aminoacylation at the organelle, leading to the incorporation of the ncAA in a site-specific and mRNA-selective manner. OT film-like organelles were previously designed to locate at four different cellular membranes—plasma membrane (PM), Golgi membrane (GM), endoplasmic reticulum membrane (ERM) and outer mitochondrial membrane (OMM); these organelles were named PMP, GMP, ERMP, and OMMP, respectively (the suffix P indicates the presence of the protein FUS (fused in sarcoma) which has a tendency to phase separate at high concentrations)[2]. The designs of the fusion proteins for the four

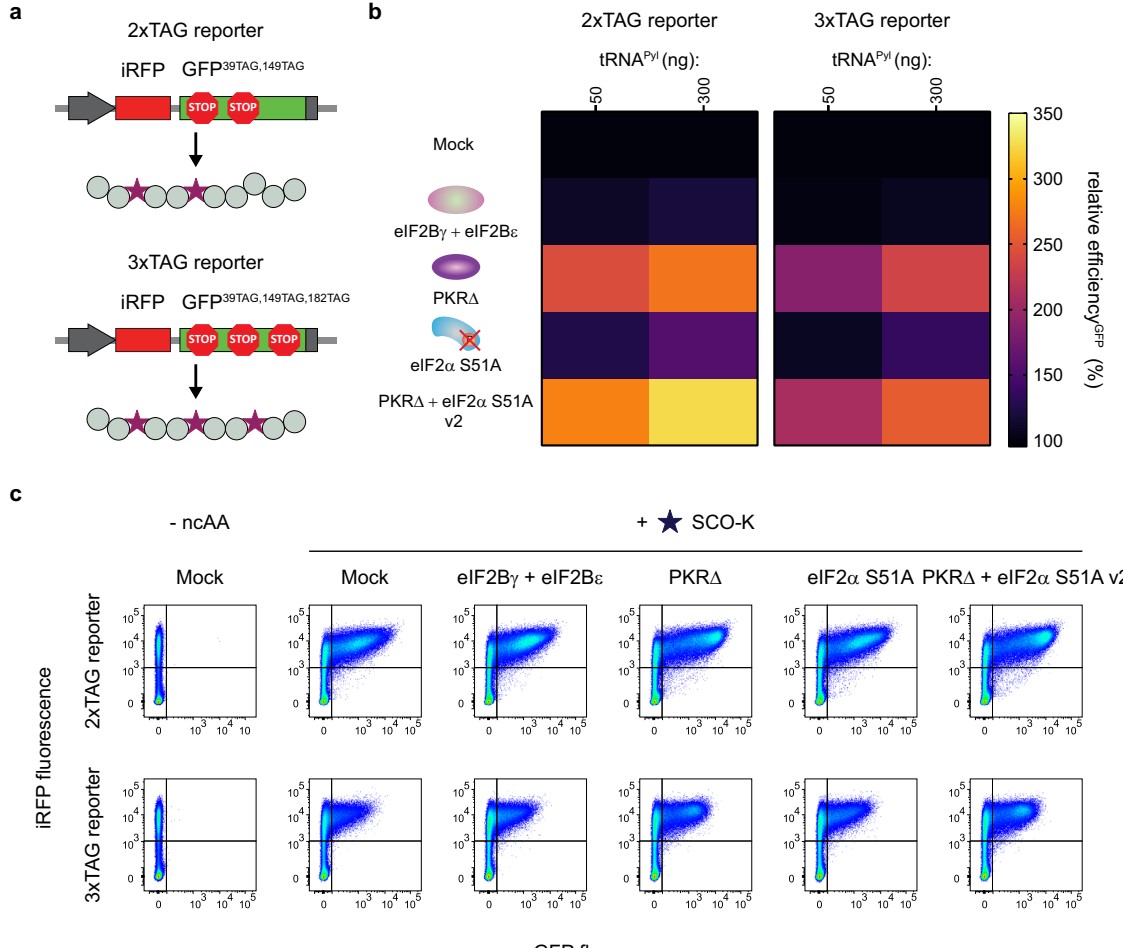

**Fig. 2 | Addition of various stress remodelers provides an increase in multiple ncAA incorporation. a** Schematic representation of iRFP-GFP[39TAG,149TAG] and iRFP-GFP[39TAG,149TAG,182TAG] reporters. GCE realization enables installation of two or three ncAAs into the corresponding reporter protein. Legend is the same as in Fig. 1a, c. **b** Heat maps illustrating the effect of adding various stress remodelers on efficiency of multiple ncAA incorporation in units of relative efficiency$^{GFP}$ (%). Percentage relative efficiency$^{GFP}$ is calculated the same way as in Fig. 1d. Median GFP signals were obtained after FC analysis of corresponding samples. The heat maps show the mean value for the relative efficiencies of three independent experiments. Source data are provided as a Source Data file. **c** FC analysis of the iRFP-GFP[39TAG,149TAG] and iRFP-GFP[39TAG,149TAG,182TAG] reporter in the absence and presence of the tested stress remodelers. Concatenated data from three independent experiments (300 ng tRNA$^{Pyl}$ plasmid) are shown.

film-like OTOs are ("-" denotes a genetic fusion, see Supplementary Note 3 for detailed information about membrane-targeting signals):

- LCK$_{1-10}$-FUS-MCP-PylRS (PMP)
- EBAG$_{1-29}$-FUS-MCP-PylRS (GMP)
- CYPIIC1$_{1-27}$-FUS-MCP-PylRS (ERMP)
- TOM20$_{1-70}$-FUS-MCP-PylRS (OMMP)

OTOs are mRNA-selective but less efficient, i.e., typically, even less ncAA modified POI is made compared with cytoplasmic PylRS[1–3]. Thus, ways to enhance such OT organelle-based GCE are urgently required. We aimed to test whether the aforementioned strategies for remodeling the cellular stress response can improve GCE efficiency for OTOs. As an OTO is not membrane-enclosed, and components are accessible from the cytoplasm, another parameter besides efficiency also becomes important, i.e., mRNA selectivity. We define mRNA selectivity as the parameter that reflects how GCE is preferentially realized for desired and ms2 loop-tagged mRNA in comparison with freely swimming untargeted mRNAs using a dual-amber fluorescent reporter (GFP$^{39TAG}$, mCherry$^{189TAG}$-ms2). In the dual-amber reporter, both GFP and mCherry mRNAs contain the amber stop codon, but only mCherry mRNA has ms2 RNA loops in the untranslated region for targeting to OTOs equipped with MCP, to which ms2 loops bind tightly. If mRNA-selective and thus protein-specific ncAA incorporation takes place, only mCherry will be produced, and a vertical population will be observed in an FC plot; if not, both GFP and mCherry will be expressed, giving a diagonal population in an FC plot (Fig. 4b). Thus, using the dual-amber reporter, two parameters for each tested GCE system can be quantified: relative efficiency$^{mCherry}$ (%)—to assess GCE level—and fold change selectivity—to estimate and quantify protein-selective ncAA incorporation. Percentage relative efficiency$^{mCherry}$ is defined as the median mCherry signal of a sample divided by the median mCherry signal of the OTO transfected with a mock plasmid (no stress remodeler added). Fold change selectivity is calculated as the median mCherry signal divided by the median GFP signal of a given system normalized to the respective ratio for the cytoplasmic control (NES-PylRS$^{AF}$) transfected with mock plasmid.

To test the OTO performance in the presence of stress remodelers we performed experiments, where we transfected the dual-amber GFP$^{39TAG}$, mCherry$^{189TAG}$-ms2 reporter, different amount of tRNA$^{Pyl}$ (10, 50, or 300 ng plasmid), OTO, and stress remodeler or mock plasmid. Supplementary Figs. 24–26 show that also for this reporter we get consistent yield enhancements compared to the reporters used in Figs. 1, 2, and 3 when using the same cytoplasmic GCE system.

**a**

ncAA-modified trastuzumab

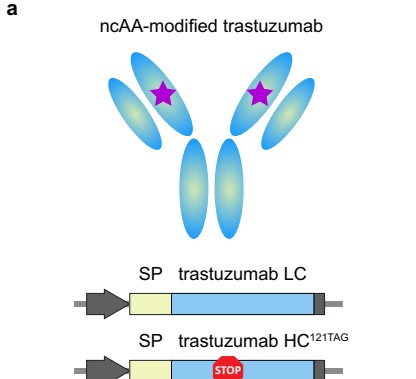

**b**

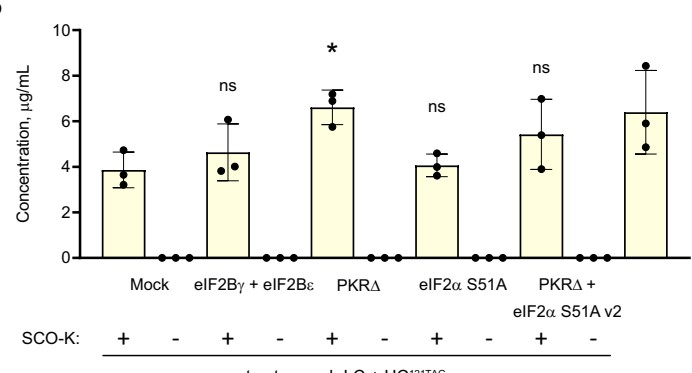

**Fig. 3 | Remodeling the cellular stress response increases the yield of ncAA-modified antibody. a** Schematic representation of ncAA-modified trastuzumab and constructs used in the study. SP denotes secretion peptide, LC - light chain, HC - heavy chain, purple star represents ncAA, gray arrow - CMV promoter; gray box - polyA signal. **b** Bar plot illustrating the effect of adding various stress remodelers on concentration of SCO-K-modified trastuzumab expressed in HEK293T cells. WT designates trastuzumab WT (trastuzumab LC + HC, without amber stop codon). Data are presented as mean values +/− standard deviation (SD). Results from three independent experiments are shown. Ns denotes not significant ($p$ value > 0.05), *$p$ value ≤ 0.05, $p$ values were calculated using one-way ANOVA with Dunnett's multiple comparison test to no stress remodeler addition (Mock). Exact $p$ values for the samples are 0.7741 (eIF2Bγ + eIF2Bε), 0.0279 (PKRΔ), 0.9974 (eIF2α S51A), 0.2585 (PKRΔ + eIF2α S51A v2). Source data are provided as a Source Data file.

For tests with OTOs, we selected the most efficient stress remodelers from strategies 2 and 3–PKRΔ and eIF2α S51A–and examined each in two conditions: (i) a simple non-anchored (NA) version using the same constructs as examined in Figs. 1, 2, and 3 and (ii) an organelle-anchored (OA) version. We hypothesize that OA constructs of stress remodelers allow to target the stress remodeler to the OTO and might help to achieve local regulatory effects. The OA version represents a fusion of organelle assemblers to the N-termini of the tested stress remodeling constructs (protein sequences of all stress remodelers used in this study are presented in Supplementary Data 1):

- LCK$_{1-10}$-FUS-PKRΔ (namely PMP-PKRΔ)
- LCK$_{1-10}$-FUS-eIF2α S51A (PMP-eIF2α S51A)
- EBAG$_{1-29}$-FUS-PKRΔ (GMP-PKRΔ)
- EBAG$_{1-29}$-FUS-eIF2α S51A (GMP-eIF2α S51A)
- CYPIIC1$_{1-27}$-FUS-PKRΔ (ERMP-PKRΔ)
- CYPIIC1$_{1-27}$-FUS-eIF2α S51A (ERMP-eIF2α S51A)
- TOM20$_{1-70}$-FUS-PKRΔ (OMMP-PKRΔ)
- TOM20$_{1-70}$-FUS-eIF2α S51A (OMMP-eIF2α S51A)

Figure 4c reports on percentage relative efficiency$^{mCherry}$ for all NA and OA constructs introduced to the OTOs using the dual-amber GFP$^{39TAG}$, mCherry$^{189TAG}$-ms2 reporter. The experiments with OT film-like organelles demonstrated that PKRΔ versions could provide a substantial (up to 1.8-fold for PMP organelle) improvement in relative efficiency$^{mCherry}$ (%) and thus in organelle-associated GCE performance (Fig. 4c, e, Supplementary Figs. 27–34, each condition with the highest fold increase in GCE efficiency for the particular OTO is indicated with a white box in Fig. 4c). For PMP and GMP organelles, application of OA-PKRΔ (PMP-PKRΔ for PMP organelle and GMP-PKRΔ for GMP organelle) gave a slightly higher effect compared with the addition of NA-PKRΔ.

Gratifyingly, the ERMP organelle performance could even achieve the level of GCE realized by the cytoplasmic PylRS (Supplementary Fig. 28). To test if OT organelle-associated translation maintains the accurate readthrough of the stop codon in the presence of PKRΔ versions, we examined the ERMP organelle introduced together with NA-PKRΔ or ERMP-PKRΔ in the absence of the ncAA SCO-K and did not observe any GFP- or mCherry-positive population in FC analysis (Supplementary Fig. 34). Importantly, the introduction of PKRΔ versions keeps or even increases fold change selectivity towards mCherry expression (Fig. 4d) for different OTOs, maintaining the key organelle property of protein-selective ncAA incorporation. Addition of NA-

eIF2α S51A to the OTOs provided low (up to 1.3-fold) increase in organelle-associated GCE, being less efficient to compare with the PKRΔ versions, while OA versions of eIF2α S51A did not enable any GCE enhancement (Supplementary Figs. 35 and 36).

We next aimed to test if anchoring the PKRΔ to the OT organelle enables local regulatory effects and enhances GCE only for the OTO, to which PKRΔ was directed, or if OA-PKRΔ provides the regulatory effect on the cytoplasm in general and thus can indirectly increase the GCE performance in distinct separately located OTO. We then repeated transfections with all four OTO and tested them in presence of PMP-PKRΔ as well as NA-PKRΔ. Somewhat unexpected, any addition of PKRΔ no matter if anchored to PMP or non-anchored led to enhanced levels of organelle-associated GCE enhancement for all tested organelles (Supplementary Figs. 37–41). This finding is inconsistent with any model in which the yield enhancing effect of PKRΔ would be assumed to originate from the unique biochemical environment inside the organelle, as we discuss further below.

We additionally examined if PKRΔ versions enable increase in organelle-associated GCE efficiency for a protein bearing multiple ncAAs. PMP organelle was transfected together with organelle-directed reporters with two or three amber sites in the reporter sequence (GFP$^{39TAG,149TAG}$-ms2 and GFP$^{39TAG,149TAG,182TAG}$-ms2, namely 2xTAG-ms2 and 3xTAG-ms2 reporter, respectively) and investigated in the presence or absence of NA-PKRΔ and PMP-PKRΔ at 300 ng tRNA$^{Pyl}$. Since the level of GFP signal was rather low we calculated two parameters to thoroughly evaluate the efficiency of PKRΔ versions: (i) percentage relative efficiency$^{GFP}$ defined as the median GFP signal of a sample divided by the median GFP signal of the sample with the corresponding PMP organelle transfected with mock plasmid, and (ii) percentage relative abundance$^{GFP}$ calculated as the percentage of GFP-positive cells in a sample divided by the percentage of GFP-positive cells in the corresponding PMP organelle sample transfected with mock plasmid. Both NA-PKRΔ and PMP-PKRΔ permitted enhancement in the GCE efficiency for GFP$^{39TAG,149TAG}$-ms2 and GFP$^{39TAG,149TAG,182TAG}$-ms2 reporters, e.g., best twofold and 3.2-fold increase in relative abundance$^{GFP}$ (%) after addition of PMP-PKRΔ for 2xTAG-ms2 and 3xTAG-ms2 reporter, respectively (Supplementary Figs. 42–44).

## Discussion

In this study, we present a method for enhancing GCE in mammalian cells via harnessing the cellular stress response to increase the rate of translation. We applied three distinct strategies (eIF2Bγ + eIF2Bε,

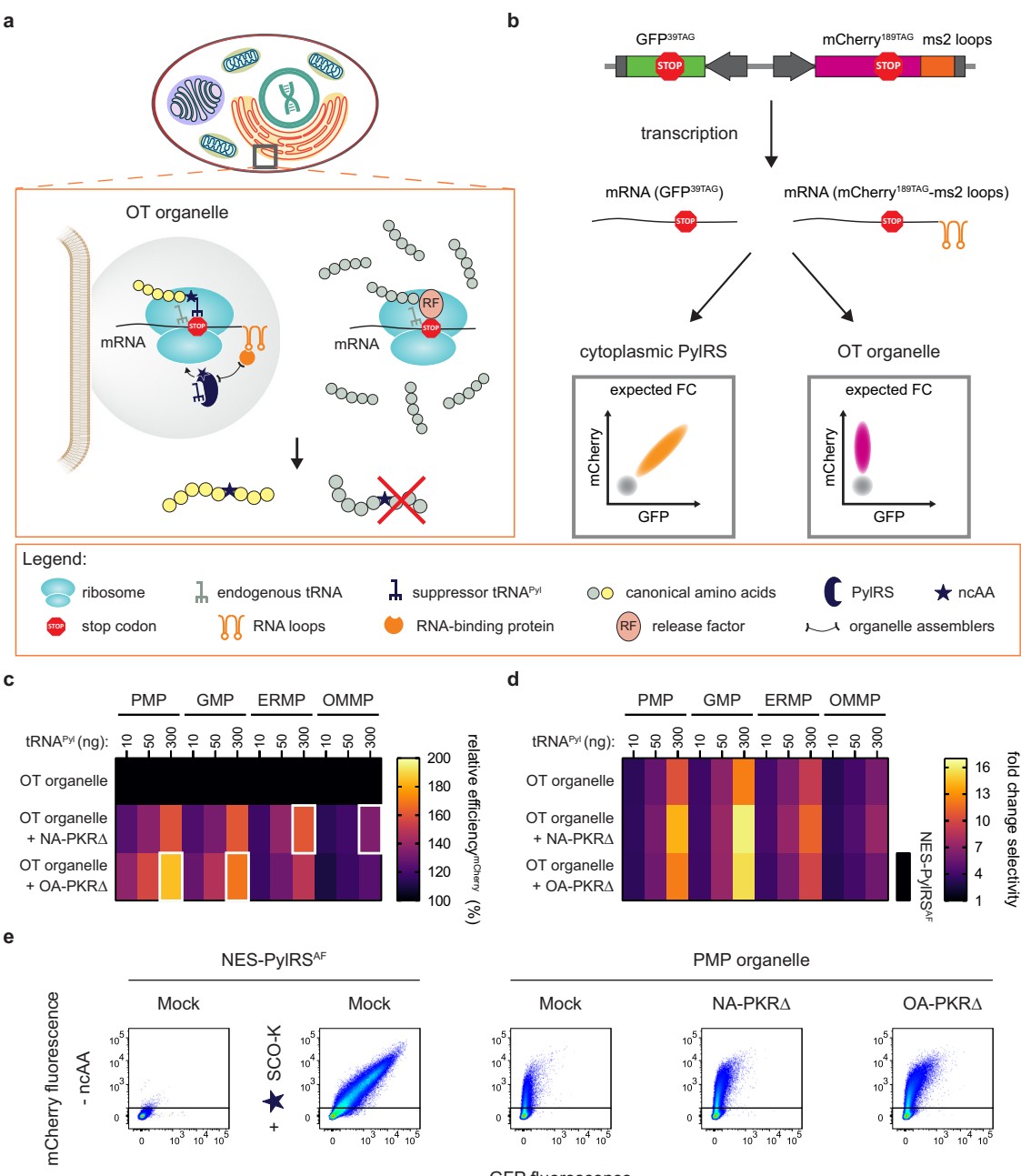

**Fig. 4 | Tailoring the PKR-dependent eIF2α phosphorylation pathway enables OTO-associated GCE enhancement. a** Schematic illustration of membrane-associated OT film-like organelles and their working principle. **b** Schematic representation of the dual-amber fluorescent reporter (GFP[39TAG], mCherry[189TAG]-ms2). The gray population represents untransfected cells; gray arrow - CMV promoter; gray box - polyA signal, FC denotes flow cytometry. **c, d** Heat maps illustrating the relative efficiency[mCherry] (%) (**c**) and fold change selectivity (**d**) of distinct OT organelles in the absence and presence of the PKRΔ variants. Percentage relative efficiency[mCherry] is calculated as the median mCherry signal of an OT organelle sample divided by the median mCherry signal for the corresponding OT organelle

transfected with mock plasmid. Fold change selectivity is quantified as the median mCherry signal divided by the median GFP signal of a given system normalized to the respective ratio for the cytoplasmic control (NES-PylRS[AF]) transfected with mock plasmid. For each organelle, conditions with the highest fold GCE improvement are outlined with a white box in Fig. 4c. The heat maps show the mean value for the relative efficiencies of three independent experiments. Source data are provided as a Source Data file. **e** FC analysis of the GFP[39TAG], mCherry[189TAG]-ms2 reporter in the absence and presence of the tested PKRΔ variants for samples equipped with the PMP organelle. Concatenated data from three independent experiments (50 ng tRNA[Pyl] plasmid) are shown.

PKRΔ, and eIF2α S51A) to affect the PKR-dependent eIF2α phosphorylation pathway and demonstrated that all three enhanced GCE. The best performing strategy for the cytoplasmic PylRS system—addition of the PKRΔ variant—provided up to 2.3-fold improvement of GCE efficiency. We then combined the two most efficient strategies (PKRΔ and eIF2α S51A) and could achieve 2.8-fold GCE enhancement. Designed stress remodelers are expected and were found to boost translational efficiency in general[18–21] and not only for amber

suppression, and thus an increase in general protein expression was monitored in parallel. Consequently, the experimental design needs to be taken into account, when using stress remodelers. However, in those applications, where increasing the yield of GCE is the primary focus, the use of remodeling the stress pathway as described in this work is of high potential.

We also examined the influence of stress remodelers' addition on multiple ncAA incorporation and could observe up to 3.2-fold GCE

enhancement for double-amber reporter (iRFP-GFP[39TAG,149TAG]) and up to 2.5-fold GCE improvement for triple-amber reporter (iRFP-GFP[39TAG,149TAG,182TAG]) with best results after addition of PKRΔ + eIF2α v2 stress remodeler. Such results demonstrate the perspective of using the stress remodelers for numerous practical applications where multiple ncAA incorporation is highly valuable (e.g., FRET studies) and for potential artificial biopolymer synthesis in mammalian cells.

Installation of diverse ncAAs into antibodies represents one of the highest potential applications of GCE technology. In this paper, we show that introduction of PKRΔ enables higher (1.7-fold) production of ncAA-modified antibody trastuzumab, which in turn is broadly exploited for ADC synthesis. We also confirmed the precise SCO-K incorporation into trastuzumab HC in mammalian cells showing that host translational machinery maintains accurate GCE-dependent readthrough of amber stop codon in the presence of PKRΔ. Our study thus shows the potential for usage of the stress remodeling strategy to enhance production of ncAA-modified antibodies in biotechnological applications, where yield of the POI is the major criterion.

We also showed that reprogramming of the cellular stress response can increase the efficiency of mRNA-selective GCE systems, such as OT film-like organelles, up to 1.8-fold for single amber suppression and up to 3.2-fold for multiple ncAA incorporation. The working principle of the OT system is to enhance local concentration, and thus local proximity between the factors that are important for amber suppression, namely the suppressor tRNA[Pyl] and the mRNA of choice. This local enrichment of GCE effectors allows to suppress preferentially the chosen stop codon inserted into the POI mRNA, so that in the rest of the cell the host translational machinery does not experience this effect. As shown previously[1,2] (e.g., when using minus MCP controls, in which case the target mRNA does not get selectively recruited to the organelle) if one of the components is no recruited to the organelle properly the system becomes less functional. In contrast, organelle-anchoring of PKRΔ resulted in a small (PMP, GMP) or no extra advantage (ERMP, OMMP) over introduction of non-anchored version of PKRΔ for single amber suppression. Results with PMP-PKRΔ show, that the effect of PKRΔ cannot be confined only to the one particular organelle environment. We speculate that such an effect can be explained by the sequestration of endogenous PKR at the site of, e.g., plasma membrane film-like compartment formed by PMP-PKRΔ due to the compulsion of dimer formation, leading to at least partial depletion of endogenous PKR out of the cytoplasm. We thus hypothesize that ultimately what matters is which system is most efficient in sequestering functional endogenous PKR to inactivate it, and the positive effect can be felt for GCE enhancement anywhere in the cell, no matter where the OT system was spatially mounted. This, however, also illustrates the benefits of membraneless organelle design, as designs of OT organelles still facilitate the communication between the organelle's microenvironment and the cytoplasm. As the film-like OT system in presence of PKRΔ shows higher efficiency and maintains its key advantage to be more selective compared with cytoplasmic GCE, our work demonstrates an approach for successful further manipulation of spatially selective protein engineering and provides a glimpse into the functioning of organelles that are inspired by the concept of phase separation.

## Methods

### Cell culture
HEK293T cells (ATCC, CRL-3216) were cultured in Dulbecco's modified Eagle's medium (DMEM, Gibco, 41965-039) supplemented with 10% FBS (Sigma−Aldrich, F7524), 1% penicillin−streptomycin (Sigma−Aldrich, P0781), 1% L-glutamine (Sigma−Aldrich, G7513), and 1% sodium pyruvate (Life Technologies, 11360) at 37 °C and 5% $CO_2$. The cells were passaged every 2–3 days up to 17 passages. The HEK293T cells were seeded 20 h prior to transient transfections (for flow cytometry and ELISA experiments) or 16 h prior to transient transfections (for Western blot analyses) in 24-well plates (Sarstedt, 83.3922) at a density of 220,000 cells/mL (500 μL/well). For GFP immunoprecipitation experiments and further LC-MS/MS analysis cells were seeded 20 h prior to transient transfections in 100 mm round cell culture dishes (CELLSTAR, Greiner Bio-One) at a density of 150,000 cells/mL (10 mL/dish).

FreeStyle™ 293-F cells (Thermo Fisher Scientific, R79007) were cultivated in FreeStyle™ 293 Expression medium accordingly to the manufactures protocol. One day before transfection the cells were split to 0.5× cells/mL.

### Constructs and cloning
Supplementary Table 1 summarizes the plasmids used in this study.

eIF2Bγ and eIF2Bε sequences were obtained by gene synthesis (Azenta Life Sciences), and the DNA sequences were codon-optimized for expression in mammalian cells. The plasmid pI.18-HA-PKR-DN was a gift from the Müller lab (EMBL, Heidelberg, Germany). The eIF2α WT sequence was acquired from the human ORF cDNA clone collection of the IMB (Institute of Molecular Biology, Mainz, Germany). eIF2Bγ and eIF2Bε sequences were inserted into a pBI vector under the CMV1 and CMV2 promoters respectively via restriction cloning. The eIF2Bγ + eIF2Bε plasmid contained both subunits under the corresponding promoters. The N-terminal PKR fragment (1–174) (PKRΔ) and eIF2α WT were cloned into a pcDNA3.1 vector using Gibson assembly. A K64E mutation was inserted into the PKRΔ using site-directed mutagenesis. An S51A mutation was introduced into the eIF2α sequence using overlap extension PCR, and the amplified fragment was then cloned into the pcDNA3.1 using Gibson assembly. An extra glycine (Gly) for (+Gly) eIF2α variants (WT and S51A) were cloned into the pcDNA3.1 using Gibson assembly. For PKRΔ + eIF2α S51A v1 stress remodelers PKRΔ and eIF2α S51A were inserted into the pBI vector under the CMV1 and CMV2 promoters respectively via restriction cloning. The plasmids PKRΔ + eIF2α S51A v2 and PKRΔ + eIF2α S51A v3 were cloned into the pcDNA3.1 using Gibson assembly. The plasmid GFP-P2A-T2A-mCherry[189TAG] (contains meGFP and mCherry M10L) was cloned using Gibson assembly into a pCI backbone. Site-directed mutagenesis was applied to generate plasmids iRFP-GFP[39TAG,149TAG] and iRFP-GFP[39TAG,149TAG,182TAG] from the iRFP-GFP[39TAG] reporter. Plasmids OA-PKRΔ and OA-eIF2α S51A were generated using Gibson assembly or overlap extension PCR respectively fusing tested stress remodelers with membrane-targeting-FUS sequences and cloning fragments into the pcDNA3.1. The heavy and the light chain of trastuzumab (trastuzumab HC and LC, respectively) equipped with serum albumin pre-proprotein secretion peptide (MKWVTFISLLFLFSSAYS) at the N-terminus were ordered as codon-optimized genes for HEK293 cells (GeneArt, Thermo Fisher Scientific) and cloned into a pcDNA3.4 vector. The trastuzumab HC[121TAG] plasmid was generated via site-directed mutagenesis. Primers used for the construction of the aforementioned plasmids are listed in the Source Data file. The plasmids NES-PylRS[AF], tRNA[Pyl], NES-PylRS[AF] + tRNA[Pyl], iRFP-GFP[39TAG], GFP[39TAG], mCherry[189TAG]-ms2, GFP[39,149TAG]-ms2, GFP[39TAG,149TAG,182TAG]-ms2 and the organelle constructs PMP, GMP, ERMP, and OMMP were published before[1,2,4,37] and directly used in this study for transient transfections.

### Transient transfections
**Transfections of HEK293T cells.** For flow cytometry, Western blot, and ELISA analyses HEK293T cells were seeded in each well of a 24-well plate, each well was transfected with a total of 1200 ng of plasmid DNA. For flow cytometry analyses for wells containing 300 ng tRNA[Pyl] plasmid, equal amounts of (1) a fluorescent reporter (iRFP-GFP[39TAG]; iRFP-GFP[39TAG,149TAG]; iRFP-GFP[39TAG,149TAG,182TAG]; GFP-P2A-T2A-mCherry[189TAG]; GFP[39TAG]; mCherry[189TAG]-ms2; GFP[39TAG,149TAG]-ms2, or GFP[39TAG,149TAG,182TAG]-ms2 reporter), (2) PylRS-containing plasmid (cytoplasmic NES-PylRS[AF] or organelle construct), (3) tRNA[Pyl] plasmid, and (4) mock plasmid

(pcDNA3.1_Zeo+) or vector with the stress remodeler, were mixed in 50 µL DMEM without phenol red. If 10 or 50 ng of tRNA$^{Pyl}$ or reporter plasmid were used, more mock plasmid was added to ensure a total plasmid mass of 1200 ng. For reporter only controls in flow cytometry experiments 300 ng of the particular reporter used for the experiment (or 50 ng/10 ng in the experiments with iRFP-GFP$^{39TAG}$ reporter titrations) were mixed with 900 ng mock plasmid (pcDNA3.1_Zeo+) (or 1150 ng/1190 ng for 50 ng/10 ng iRFP-GFP$^{39TAG}$ reporter, respectively) in 50 µL DMEM without phenol red. For entire mock transfections in flow cytometry experiments 1200 ng mock plasmid (pcDNA3.1_Zeo+) were added to 50 µL DMEM without phenol red. For Western blot analyses of the ISR two sets of transfection conditions were tested: (1) 300 ng PylRS-containing plasmid (cytoplasmic NES-PylRS$^{AF}$), 300 ng tRNA$^{Pyl}$ plasmid, and 600 ng mock plasmid (pcDNA3.1_Zeo+) or vector with the tested stress remodeler were mixed in 50 µL jetPRIME® buffer (Supplementary Figs. 16, 49) or (2) 1200 ng stress remodeler was added to 50 µL jetPRIME® buffer (Supplementary Figs. 17, 50). For entire mock transfections in Western blot experiments 1200 ng mock plasmid (pcDNA3.1_Zeo+) were added to 50 µL jetPRIME® buffer. For trastuzumab expression and further ELISA measurements equal amounts of (1) trastuzumab LC plasmid, (2) trastuzumab HC or trastuzumab HC$^{121TAG}$ plasmid, (3) NES-PylRS$^{AF}$ + tRNA$^{Pyl}$ plasmid, and (4) mock plasmid (pcDNA3.1_Zeo+) or vector with the tested stress remodeler, were mixed in 50 µL DMEM without phenol red. For LC-MS/MS analysis each 100 mm cell culture dish was transfected with a total of 24 µg of plasmid DNA. Equal amounts of (1) a fluorescent reporter (iRFP-GFP$^{39TAG}$), (2) PylRS-containing plasmid (cytoplasmic NES-PylRS$^{AF}$), (3) tRNA$^{Pyl}$ plasmid, and (4) mock plasmid (pcDNA3.1_Zeo+) or vector with the stress remodeler, were mixed in 1 mL DMEM without phenol red. Polyethylenimine (PEI, 1 µg/µL, Sigma, 408727-100 ml) was added to all transfections at a DNA/PEI ratio of 1:3 for performing flow cytometry, LC-MS/MS, and ELISA analyses. jetPRIME® transfection reagent (Polyplus, 101000046) was introduced to all transfections at a DNA/transfection reagent ratio of 1:2 for conducting Western blot analysis of the ISR status. The mixtures were vortexed for 10 s, spun for 5 s, and incubated for 15 min (for samples with PEI) or 10 min (for samples with jetPRIME® transfection reagent) at room temperature before being added dropwise into the wells or dishes. After 4 h, the medium in each well or dish was removed and replaced with fresh medium buffered with 25 mM HEPES (pH 7.25). Samples incubated in the presence of ncAA contained 250 µM cyclooctyne-lysine (SCO-K, SiChem, SC-8000) in the buffered medium.

**Transfections of FreeStyle™ 293-F cells.** For each transfection 20 mL of FreeStyle™ 293-F cells were prepared at a cell density of $1 \times 10^6$ cells/mL. 20 µg DNA were used for each transfection using four different plasmids in equal amounts: (1) trastuzumab LC plasmid, (2) trastuzumab HC$^{121TAG}$ or trastuzumab HC plasmid, (3) NES-PylRS$^{AF}$ + tRNA$^{Pyl}$ plasmid, and (4) mock plasmid (pcDNA3.1_Zeo+) or pl.18-HA-PKR-DN (PKRΔ). The plasmids were pipetted into 2 mL OptiPRO™ SFM medium (Thermo Fisher Scientific, 12309019), mixed with 80 µL PEI MAX (Polysciences, 24765-100, 1 mg/mL stock prepared following manufactures instruction), vortexed 3 × 5 s and incubated for 10 min at RT before added into the cell suspension. The expression was incubated for 6 days at 37 °C, 8% CO$_2$ at 120 rpm (50 mm shaking throw). Samples were incubated in the presence of 500 µM cyclooctyne-lysine (SCO-K, SiChem, SC-8000) in the medium buffered with 25 mM HEPES (pH 7.25).

**Flow cytometry and data analysis**
HEK293T cells were incubated for 24 h after changing the medium and then prepared for FC analysis. Cells were rinsed once with 200 µL of 1× PBS, detached using 100 µL of trypsin-EDTA (0.05%, with phenol red, Gibco, 25300-054) and incubated for 5 min at 37 °C and 5% CO$_2$. Next, 900 µL of resuspension buffer 1 (1× PBS containing 10% FBS, 2 mM sodium azide, and 2 mM EDTA) was added to each well, and the cells

were collected on ice, transferred to 1.5 mL tubes, and centrifuged for 5 min at 400 × g and 4 °C. The supernatants were discarded, and the pellets were washed with 900 µL of resuspension buffer 2 (1× PBS containing 3% bovine serum albumin (BSA), 2 mM sodium azide, and 2 mM EDTA). Then, the cells were again centrifuged for 5 min at 400 × g and 4 °C. The supernatants were discarded, the cell pellets were resuspended in 250 µL of resuspension buffer 2, and 300 µL of cell suspension was taken for FC analysis. 3 µL of DAPI (50 µg/mL) were added to the cell suspension for live cell staining, followed by incubation of the samples for 1–2 min on ice before measurement. Data acquisition was performed with an LSRFortessa Cell Analyzer (BD Biosciences). Analysis was performed using FlowJo version 10.7.1 (BD Biosciences). First, the population of HEK293T cells was gated (using FSC-A x SSC-A parameters), and then a single cell population was selected (SSC-W x SSC-A). Next, live cells were picked (SSC-W x 405–450/50 channel) (Supplementary Figs. 45–48). At least 100,000 live cells were collected and analyzed per measured sample. The next steps depended on the type of reporter used in the experiment:

**FC analysis of iRFP-GFP$^{39TAG}$, iRFP-GFP$^{39TAG,149TAG}$ and iRFP-GFP$^{39TAG,149TAG,182TAG}$ reporters.** The fluorescence of GFP was acquired using a 488 nm laser and a 530/30 bandpass filter. For detecting iRFP, a 640 nm laser with a 730/45 bandpass filter was used. Measurements of live cells were plotted with GFP fluorescence on the OX axis and iRFP fluorescence on the OY axis. The plot was divided into four quadrants. The workflow of cell gating is shown in Supplementary Fig. 45. To calculate median GFP signal, only cells from the top-right quadrant (Q2) were considered. For quantification of median iRFP signal cells from the two top quadrants (top left and top right; Q1 and Q2, respectively) were selected. Microsoft Excel files containing median iRFP and GFP values were generated from the FlowJo workspace and used for calculating relative efficiency$^{GFP}$ (%) or relative efficiency$^{iRFP}$ (%). Percentage relative efficiency$^{GFP}$ or percentage relative efficiency$^{iRFP}$ was calculated as the median GFP or iRFP signal for each particular sample divided by the median GFP or iRFP signal for samples with the addition of mock plasmid. Three biological replicates were performed for all experiments except for the experiment with titrated iRFP-GFP$^{39TAG}$ reporter, which was performed once. Values for the relative efficiencies were calculated in Microsoft Excel and transferred to Prism software (GraphPad). Mean values and standard deviation were quantified if three replicates were performed for the experiment and then the corresponding heat maps and bar plots were generated. For the experiment with titrated iRFP-GFP$^{39TAG}$ reporter the relative efficiencies were calculated in Microsoft Excel and transferred to Prism software (GraphPad) to generate the corresponding line charts.

**FC analysis of the GFP-P2A-T2A-mCherry$^{189TAG}$ reporter.** The fluorescence of GFP was acquired using a 488 nm laser and a 530/30 bandpass filter. For detecting mCherry, a 561 nm laser with a 610/20 bandpass filter was used. Measurements of live cells were plotted with GFP fluorescence on the OX axis and mCherry fluorescence on the OY axis. The plot was divided into four quadrants. The workflow of cell gating is shown in Supplementary Fig. 46. To calculate median GFP signal, only cells from the top-right quadrant (Q2) were considered. For quantification of median mCherry signal cells from the two right quadrants (top right and bottom right; Q2 and Q3, respectively) were selected. Microsoft Excel files containing median mCherry and GFP values were generated from the FlowJo workspace and used for calculating relative efficiency$^{GFP}$ (%) or relative efficiency$^{mCherry}$ (%). Percentage relative efficiency$^{GFP}$ or percentage relative efficiency$^{mCherry}$ was calculated as the median GFP or mCherry signal for each particular sample divided by the median GFP or mCherry signal for samples with the addition of mock plasmid. One experiment was performed for all

tested stress remodelers. The relative efficiencies were calculated in Microsoft Excel and transferred to Prism software (GraphPad) to generate the corresponding bar plots.

**FC analysis of the dual-amber fluorescent reporter (GFP$^{39TAG}$, mCherry$^{189TAG}$-ms2).** The fluorescence of GFP was acquired using a 488 nm laser and a 530/30 bandpass filter. For detecting mCherry, a 561 nm laser with a 610/20 bandpass filter was used. Measurements of live cells were plotted with GFP fluorescence on the OX axis and mCherry fluorescence on the OY axis. The plot was divided into two quadrants. The workflow of cell gating is shown in Supplementary Fig. 47. To calculate median/mean mCherry signal and median/mean GFP signal, only cells from the top half were selected. Microsoft Excel files containing median/mean mCherry and median/mean GFP values were generated from the FlowJo workspace and used for calculating relative efficiency$^{mCherry}$ (%), relative efficiency$^{GFP}$ (%) and fold change selectivity. Percentage relative efficiency$^{mCherry}$ was calculated as the median mCherry signal of a NES-PylRS$^{AF}$ or OT organelle sample divided by the median mCherry signal for the NES-PylRS$^{AF}$ or corresponding OT organelle respectively transfected with mock plasmid. For Supplementary Figs. 28, 29, and 36 relative efficiency$^{mCherry}$ (%) was calculated as the median/mean mCherry signal of a sample divided by the median/mean mCherry signal for cytoplasmic control (NES-PylRS$^{AF}$) transfected with mock plasmid. Percentage relative efficiency$^{GFP}$ was calculated as the median GFP signal of a NES-PylRS$^{AF}$ or OT organelle sample divided by the median GFP signal for the NES-PylRS$^{AF}$ or corresponding OT organelle respectively transfected with mock plasmid. Fold change selectivity was quantified as the median/mean mCherry signal divided by the median/mean GFP signal of a given system normalized to the respective ratio for the cytoplasmic control (NES-PylRS$^{AF}$) transfected with mock plasmid. Three biological replicates were performed for all experiments except for the experiment where PMP-PKRΔ was tested with differently localized OTOs, which was performed once. Values for the percentage relative efficiencies and the fold change selectivity were calculated in Microsoft Excel and transferred to Prism software (GraphPad). Mean values and standard deviation were quantified if three replicates were performed for the experiment and then the corresponding heat maps and bar plots were generated. For the experiment with PMP-PKRΔ and differently localized OT organelles, the relative efficiencies and the fold change selectivity were calculated in Microsoft Excel and transferred to Prism software (GraphPad) to generate the corresponding heat maps and bar plots.

**FC analysis of GFP$^{39TAG,149TAG}$-ms2 (2xTAG-ms2) and GFP$^{39TAG,149TAG,182TAG}$-ms2 (3xTAG-ms2) reporters.** The fluorescence of GFP was acquired using a 488 nm laser and a 530/30 bandpass filter. Measurements of live cells were plotted with GFP fluorescence on the OX axis and SSC-W on the OY axis. The plot was divided into two quadrants. The workflow of cell gating is shown in Supplementary Fig. 48. To calculate median GFP signal and to obtain percentage of GFP+ cells, only cells from the right quadrant were selected. Microsoft Excel files containing median GFP values and percentages of GFP+ cells were generated from the FlowJo workspace and used for calculating relative efficiency$^{GFP}$ (%) and relative abundance$^{GFP}$ (%). Percentage relative efficiency$^{GFP}$ was calculated as the median GFP signal of OT (PMP) organelle sample divided by the median GFP signal for the corresponding PMP organelle transfected with mock plasmid. Percentage relative abundance$^{GFP}$ was calculated as percentage of GFP+ cells in a sample divided by percentage of GFP+ cells in the corresponding OT organelle sample transfected with mock plasmid. One experiment was performed for all tested stress remodelers. The relative efficiency and the relative abundance were calculated in Microsoft Excel and transferred to Prism software (GraphPad) to generate the corresponding bar plots.

**Western blot analysis of the ISR status**

One hour before harvesting, sodium arsenite (at 0.5 mM final concentration) was added to the separate untransfected samples for positive control of the ISR. Sterile water was used for samples with negative control of the stress inducer addition. For 30 min before harvesting all samples (except negative controls where sterile water was added) were treated with 20 μg/mL puromycin (Gibco, A1113803) enabling puromycin incorporation assay. 24 h after changing medium, cells were washed once with 200 μL 1× PBS and then lysed using 60 μL radioimmunoprecipitation assay (RIPA) buffer (150 mM NaCl, 1.0% Triton X-100, 0.5% sodium deoxycholate, 0.1% SDS, 50 mM Tris, pH 8.0) supplemented with cOmplete Protease Inhibitor Cocktail (Roche, 11873580001), PhosSTOP (phosphatase inhibitor, Roche, 04906845001), 1 mM magnesium chloride, and Sm nuclease (Protein Production Core Facility, IMB Mainz). Cells were incubated on ice for 20 min (with pipetting every 10 min), then protein concentration in each sample was measured using Pierce™ 660 nm Protein Assay Kit (Thermo Fisher Scientific). Protein samples were normalized to have equal protein mass in each sample and were subjected to SDS-PAGE [NuPAGE 4–12% Bis-Tris-Gel run in NuPAGE MOPS SDS Running Buffer (Thermo Fisher Scientific)]. Proteins were transferred to nitrocellulose membranes (Trans-Blot Turbo Midi 0.2 μm Nitrocellulose Transfer Packs, Bio-Rad, 1704159) using Trans-Blot Turbo Transfer System (Bio-Rad). The membranes were blocked with 2% BSA in 1× TBST for 60 minutes at room temperature and then incubated with primary antibodies dependent on the performed assay: for evaluation of level of phosphorylated eIF2α rabbit anti-phospho-eIF2α antibody (Cell Signaling, 3398) and mouse anti-total eIF2α antibody (Thermo Fisher Scientific, AHO0802) were used both in dilution 1:1000 in 2% BSA in 1× TBST, while for puromycin incorporation assay mouse anti-puromycin antibody (Merck Millipore, clone 12D10, MABE343) was applied in dilution 1:1000 in 2% BSA in 1× TBST. Membranes were incubated with primary antibodies overnight at 4 °C. Next day membranes were washed three times with 2% BSA in 1× TBST and incubated with secondary antibodies: IRDye® 800CW goat anti-rabbit antibody (LI-COR, 926-32211, 1:10,000 in 2% BSA in 1× TBST) and IRDye® 680RD goat anti-mouse antibody (LI-COR, 926-68070, 1:10000 in 2% BSA in 1× TBST)−for assessment of eIF2α phosphorylation level−and IRDye® 800CW goat anti-mouse antibody (LI-COR, 925-32210, 1:10,000 in 2% BSA in 1× TBST)−for puromycin incorporation assay. Secondary antibodies were incubated for 1 h at room temperature. Then membranes were washed three times with 2% BSA in 1× TBST and scanned with Odyssey M Imaging System (LI-COR). After that, membranes were reprobed with primary antibodies for loading control: mouse anti-cyclophilin B antibody (Abcam, clone CL3901, ab236760, 1:1000 in 2% BSA in 1× TBST) was used for the experiment with evaluation of phosphorylated eIF2α level, and rabbit anti-cyclophilin B antibody (Invitrogen, PA1-027A, 1:1000 in 2% BSA in 1× TBST) was applied for the membranes used for puromycin incorporation assay. Primary antibodies were incubated overnight at 4 °C. Next day membranes were washed three times with 2% BSA in 1× TBST and incubated with secondary antibodies: IRDye® 680RD goat anti-mouse antibody (LI-COR, 926-68070, 1:10,000 in 2% BSA in 1× TBST)−for the membranes with assessment of eIF2α phosphorylation level−and IRDye® 680RD goat anti-rabbit antibody (LI-COR, 926-68071, 1:10000 in 2% BSA in 1× TBST)−for the membranes with puromycin incorporation assay. Secondary antibodies were incubated for 1 h at room temperature. Then membranes were washed three times with 2% BSA in 1× TBST and scanned with Odyssey M Imaging System (LI-COR). Quantitative analysis of signals from membranes was performed in Image Studio Lite 5.2 software, normalized ratios were calculated in Microsoft Excel, and Prism software (GraphPad) was used to generate the corresponding bar plots.

## ELISA measurements and data analysis

The supernatants obtained from transfected HEK293T cells were sterile filtered through a 0.22 μm syringe filter. The cleared supernatants from the expressions were used for ELISA analysis. For the detection of trastuzumab antibody the IgG (Total) Human Uncoated ELISA Kit (Thermo Fisher Scientific, 88-50550-88) was used. The ELISA analysis was performed following the manufacturer's instructions. The samples were prediluted 1:100 in assay Buffer A, followed by 2 serial dilutions of 1:2. Tecan Infinite M200 Pro was used to measure absorbance values at 450 nm and 570 nm. Microsoft Excel files were generated after the measurements, and background-corrected values were calculated according to the manufacturer's instructions. Then the corrected values were analyzed with 5PL (five parameter logistic) curve mode using online data analysis tool MyAssays (https://www.myassays.com/). Mean concentration for a sample in one experiment was calculated using obtained concentration values for serial dilutions. Three independent experiments were performed for all tested samples. The mean concentrations for each sample from each experiment was downloaded into Prism software (GraphPad), and average (mean) values and SD for samples' concentrations were quantified and applied to generate the corresponding bar plot.

## Purification of trastuzumab[121SCO-K] and trastuzumab WT

The purification of the cleared supernatants obtained after expression in FreeStyle™ 293-F cells were conducted with a HiTrap Fibro™ PrismA column (Cytiva, 17549855). 2 mL of 10x PBS was added to the 20 mL of cleared supernatants. Then the samples were loaded on the column, which was equilibrated in 1× PBS. After washing with 20 mL of 1× PBS, the samples were eluted in a 10 mL gradient using 0.1 M sodium acetate pH 3.4 as elution buffer. Fractions were collected during the elution step and analyzed by SDS-PAGE. The trastuzumab containing fractions were pooled and concentrated in a 30 kDa cut-off Amicon-ultra centrifugal filter (Millipore, UFC903024).

## Mass spectrometry analyses

**Purification of iRFP-GFP[39SCO-K] from HEK293T cells.** GFP immunoprecipitation experiments were performed after 24 h incubation with SCO-K. Cells were washed with 1× PBS, detached using 1 mL 0.05% trypsin-EDTA (with phenol red, Gibco, 25300-054), and resuspended in 9 mL medium. Then cells were transferred to a 10 mL tube and centrifuged for 20 min at $500 \times g$, 20 °C. The supernatants were discarded, and cells were resuspended in 1 mL of 1× PBS and transferred to a 1.5 mL tube, then they were pelleted for 5 minutes at $500 \times g$, 4 °C. The supernatants were discarded, and cells were lysed using 200 μL RIPA buffer (150 mM NaCl, 1.0% Triton X-100, 0.5% sodium deoxycholate, 0.1% SDS, 50 mM Tris, pH 8.0) supplemented with cOmplete Protease Inhibitor Cocktail (Roche, 11873580001), 1 mM phenylmethylsulfonyl fluoride (PMSF, protease inhibitor), 1 mM magnesium chloride, and Sm nuclease (Protein Production Core Facility, IMB Mainz). Cells were incubated on ice for 20 min (with pipetting every 10 min), and then iRFP-GFP[39SCO-K] was purified using GFP-Trap Magnetic Agarose Particles M-270 (ChromoTek, gtdk-20). 25 mL slurry of GFP beads were added to 500 mL ice-cold wash buffer (10 mM Tris/Cl (pH 7.5), 150 mM NaCl, 0.05% Nonidet™ P40 Substitute, 0.5 mM EDTA) and then magnetically separated. The lysates were diluted with 300 mL dilution buffer (10 mM Tris/Cl (pH 7.5), 150 mM NaCl, 0.5 mM EDTA) and added to the beads for one hour incubation at 4 °C on a rolling shaker. Then the beads were magnetically separated, the supernatants were removed, and beads were washed three times with 500 μL wash buffer. Subsequently, the proteins were eluted using 40 μL 2× SDS-sample (Laemmli) buffer (120 mM Tris/Cl pH 6.8, 20% glycerol, 4% SDS, 0.04% bromophenol blue, 10%

β-mercaptoethanol). Then the protein samples were heated up at 95 °C for 5 min and subsequently diluted with 40 μL distilled H₂O. 20 μL of protein samples were separated via SDS-PAGE [NuPAGE 4%–12% Bis-Tris-Gel run in NuPAGE MOPS SDS Running Buffer (Thermo Fisher Scientific)]. Subsequently, gels were stained with Coomassie (Quick Coomassie Stain, Serva, 35081.01). The corresponding bands were cut out of the gel and stored in 1.5 mL tubes at −20 °C until further LC-MS/MS analysis.

**LC-MS/MS analysis (peptide mass fingerprinting) of iRFP-GFP[39SCO-K].** Coomassie-stained protein gel bands were cut into small cubes, followed by destaining in 50% ethanol/25 mM ammonium bicarbonate. Afterwards, proteins were reduced in 10 mM DTT at 56 °C and alkylated by 50 mM iodoacetamide in the dark at room temperature. Enzymatic digestion of proteins was performed using trypsin (1 μg per sample) in 50 mM ammonium bicarbonate overnight at 37 °C. Following peptide extraction sequentially using 30% and 100% acetonitrile, the sample volume was reduced in a centrifugal evaporator to remove residual acetonitrile. The resultant peptide solution was purified by solid phase extraction in C₁₈ StageTips[38].

Peptides were separated in an in-house packed 30-cm analytical column (inner diameter: 75 μm; ReproSil-Pur 120 C₁₈-AQ 1.9-μm beads, Dr. Maisch GmbH) by online reverse phase liquid chromatography through a 105-min non-linear gradient of 1.6–32% acetonitrile with 0.1% formic acid at a nanoflow rate of 225 nL/min. The eluted peptides were sprayed directly by electrospray ionization into a Q Exactive Plus Orbitrap mass spectrometer (Thermo Fischer Scientific). Mass spectrometry measurement was conducted in data-dependent acquisition mode using a top10 method with one full scan (mass range: 300–1650 $m/z$; resolution: 70,000, target value: $3 \times 10^6$, maximum injection time: 20 ms) followed by 10 fragmentation scans via higher energy collision dissociation (HCD; normalized collision energy: 25%, resolution: 17,500, target value: $1 \times 10^5$, maximum injection time: 120 ms, isolation window: 1.8 $m/z$). Precursor ions of unassigned or +1 charge state were rejected. In addition, precursor ions already isolated for fragmentation were dynamically excluded for 20 s.

Raw data files were processed by MaxQuant software (version 1.6.5.0)[39] using its built-in Andromeda search engine[40]. Spectral data were searched against a target-decoy database consisting of the forward and reverse sequences of the iRFP-GFP reporter protein variants whose residue position 39 of its GFP sequence is the SCO-K (O as a place holder) or any one of the 20 natural amino acids, the UniProt human reference proteome (release 2022_03; 101,761 entries), and a list of 246 common contaminants. Trypsin/P specificity was assigned. Carbamidomethylation of cysteine and pyrrolysine to SCO-K substitution were set as fixed modification. Oxidation of methionine and acetylation of the protein N-terminus were chosen as variable modifications. A maximum of 2 missed cleavages were tolerated. The minimum peptide length was set to be 7 amino acids. False discovery rate (FDR) was set to 1% for both peptide and protein identifications. The processed MS/MS spectra were visualized and annotated in the Viewer module of MaxQuant[41].

**Determination of intact mass of trastuzumab[121SCO-K] and trastuzumab WT.** 50 μg of trastuzumab[121SCO-K] or trastuzumab WT samples were deglycosylated using the immobilized PNGase F kit (Genovis, G1-PF6-010) to remove the N-glycans. After eluting from the spin column of the kit, the samples were concentrated in a 30 kDa Amicon Ultra-0.5 Centrifugal Filter Unit (Millipore, UFC503096), and the concentration was measured by spectrophotometry (DeNovix, DS-11, using absorbance at 280 nm). 100 mM DTT was added to the samples prior mass spectrometry analysis.

The intact mass for each sample was measured using a Q-Tof Premier (Waters) (Proteomics core facility, EMBL Heidelberg). Protein samples were acidified using 1% TFA prior to injection onto the Acquity UPLC System (Waters Corporation). Approximately 2 μg of each sample were loaded onto a protein separation column (Acquity UPLC Protein BEH C4 column, 2.1 mm × 150 mm, 1.7 um). The outlet of the analytical column was coupled directly to a quadrupole time of flight (Q-TOF) Premier mass spectrometer (Waters/Micromass) using the standard ESI source in positive ion mode. Solvent A was water, 0.1% formic acid, and solvent B was acetonitrile, 0.1% formic acid. For the Q-TOF, a spray voltage of 3.5 kV was applied with a cone voltage of 35 V and extraction cone at 5 V. The desolvation temperature was set at 350 °C, with source temperature 120 °C. Desolvation gas was nitrogen at a flow rate of 400 L/h. Data was acquired in continuum mode over the mass range 500–3500 $m/z$ with a scan time of 1 s and an interscan delay of 0.1 s. Data were externally calibrated against an intact protein reference standard (Thermo Fisher Scientific), acquired immediately prior to sample data acquisition. Spectra from the chromatogram protein peak were then summed, and intact mass was calculated using the MaxEnt1 maximum entropy algorithm (Waters/Micromass) to give the zero charge deconvoluted molecular weight.

## Statistics and reproducibility

For each replicate of FC analysis for each sample at least 100000 single live HEK293T cells were collected to ensure that coefficient of variation even for target populations with relatively low frequency (~1%) is smaller than 5%. In all experiments for all samples the frequency of target population was >1%. For ELISA and Western blot analysis no statistical method was used to predetermine sample size; three (for ELISA) and five (for Western blot) independent experiments were performed to obtain enough data for statistical analysis. No data were excluded from the analyses. The HEK293T cells were identically seeded into wells, then a well was randomly chosen for transfection with particular plasmid set and later all samples were prepared for FC, ELISA, or Western blot analysis. The investigators were blinded to group allocation during data collection and analysis. All calculations of standard deviations presented on the bar plots as well as statistical significance tests were performed in Prism software (GraphPad). Adjusted $p$ values for Supplementary Fig. 1 were calculated using Benjamini-Hochberg multiple comparisons correction in RStudio. Next, statistical significance tests were carried out for datasets presented in Fig. 3 and Supplementary Figs.:

- for Fig. 3 values for the concentration of antibodies obtained for samples with implemented stress remodelers were compared with antibody concentration measured for sample with no stress remodeler addition using one-way ANOVA with Dunnett's multiple comparison test.
- for Supplementary Fig. 1 all relative efficiency$^{GFP}$ (%) values obtained after testing stress remodelers were compared with relative efficiency$^{GFP}$ (%) for no stress remodeler addition (equal to 100) using one-sample (two-tailed) $t$-test with Benjamini-Hochberg multiple comparisons correction (false discovery rate = 0.05). Comparisons were made inside the group of values obtained at the same tRNA$^{Pyl}$ level.
- for Supplementary Fig. 16 normalized ratios for tested samples were compared with normalized ratios for Mock + ncAA condition (no stress remodeler added to GCE machinery, equal to 1) using one-sample (two-tailed) $t$-test.
- for Supplementary Fig. 18 relative efficiency$^{GFP}$ (%) values were compared with relative efficiency$^{GFP}$ (%) for the sample with no introduced stress remodeler (equal to 100) using one-sample (two-tailed) $t$-test. Comparisons were made inside the group of values obtained with the same reporter (the double-amber or the triple-amber reporter) and at the same tRNA$^{Pyl}$ level.
- For Supplementary Fig. 24 relative efficiency$^{GFP}$ (%) and relative

efficiency$^{mCherry}$ (%) values were compared with relative efficiency$^{GFP}$ (%) or relative efficiency$^{mCherry}$ (%) respectively for the condition with no added stress remodeler (equal to 100) using one-sample (two-tailed) $t$-test. Comparisons were made inside the group of relative efficiency$^{GFP}$ (%) or relative efficiency$^{mCherry}$ (%) values obtained at the same tRNA$^{Pyl}$ level.

- for Supplementary Figs. 27 and 35 relative efficiency$^{mCherry}$ (%) values were compared with relative efficiency$^{mCherry}$ (%) for the same OT organelle with no introduced stress remodeler (equal to 100) using one-sample (two-tailed) $t$-test. Comparisons were made inside the group of values obtained for the particular OT organelle (PMP, GMP, ERMP or OMMP) and at the same tRNA$^{Pyl}$ level.
- for Supplementary Figs. 28 and 36 relative efficiency$^{mCherry}$ (%) values were compared with relative efficiency$^{mCherry}$ (%) for the same OT organelle with no introduced stress remodeler and normalized to cytoplasmic GCE level using one-way ANOVA with Dunnett's multiple comparison test. Comparisons were made inside the group of values obtained for the particular OT organelle (PMP, GMP, ERMP or OMMP) and at the same tRNA$^{Pyl}$ level.

Exact $p$ values are provided in the Source Data file. For FC analysis, we stress that the median/mean were calculated from over 100,000 live cells, which gives high accuracy even for a single data set. We note that statistical tests are defined for large sample numbers and need to be used with caution with low sample numbers. However, our findings are consistent across a large number of assays and conditions in total amounting to dozens of experiments.

## Reporting summary

Further information on research design is available in the Nature Portfolio Reporting Summary linked to this article.

## Data availability

The data generated in this study are provided in the main text and Supplementary Information. Source data are provided with this paper.

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

## Acknowledgements

We thank Christopher D. Reinkemeier and all the members of the Lemke laboratory for helpful discussions. We thank the Flow Cytometry Core Facility, Proteomics Core Facility at the Institute of Molecular Biology (IMB) and Proteomics Core Facility at the European Molecular Biology Laboratory (EMBL) for expert assistance. We thank the Müller lab (EMBL, Heidelberg, Germany) for the gift of plasmid pl.18-HA-PKR-DN. E.A.L. acknowledges funding from the Volkswagenstiftung (Life).

## Author contributions

M.E.S., C.K., and E.A.L. conceived the project. M.E.S. and C.K. performed the experiments. M.E.S., C.K., and E.A.L. analyzed the data and co-wrote the manuscript.

## Funding

## Competing interests

E.A.L. and C.K. have submitted a patent application to the European Patent Office pertaining to the aspect of this work of the incapability of PKR of phosphorylating eIF2α and therefore enhancing the expression level (application number EP20158737A). M.E.S. declares no competing interests.
