## [Peer Review File · Nature Communications]

REVIEWER COMMENTS

Reviewer #1 (Remarks to the Author):

Sushkin et al., investigated the effects of intervening in the regulation of the protein kinase R (PKR)-dependent eIF2 α phosphorylation pathway, which according to the authors, is a stress response that occurs in mammalian cells during presence of heightened RNA levels (among other cellular stressors) such as those that may arise during genetic code expansion (GCE), which lowers translational initiation. To this effect, the authors explored 3 possible strategies to perturbing this pathway to the benefit of GCE: 1) introducing a binary eIF2Bye domain to catalyze GTP replacement in the eIF2 complex and thereby increase translation initiation; 2) introducing a truncated form of PKR (PKR Δ) that limits eIF2 inhibition; and 3) introducing a mutant form of eIF2 α (eIF2 α -S51A) that is resistant to phosphorylation (and therefore inhibition). Using an iRFP-TAG-GFP reporter system, the authors observed that the best intervention improved suppression efficiency and GFP fluorescence by up to 2.2-fold. The PKR Δ (at lower tRNA levels) having the most substantial effect, followed by eIF2 α -S51A. The authors then employ these two strategies to improve GCE efficiency on orthogonal, phase transitioned organelles. They observed that PKR Δ improved efficiency within organelles to a level approaching those of where GCE components were not targeted to orthogonal organelles, while improving selectivity. Interestingly, targeting the PKR Δ protein to these orthogonal organelles did not as markedly improve suppression efficiency as much as cytoplasmic PKR Δ . While targeting the stress response is an innovative approach to improve GCE efficiency the level of improvement was minimal. Targeting phase transition organelles is interesting but it is unclear from this application how this will advance protein engineering or cell biology. The minimal improvement in GCE demonstrated here will likely have a modest impact on the study of cellular function.

Major Comments:

1. The authors have developed a tool that improves the efficiency of GCE marginally, and nearly restores the efficiency deficit encountered through spatially orthogonalizing GCE-dependent translation to phase-transitioned organelles. These two experiments, lack the transformative impact to advance the field. Nevertheless, the quality of the science is sound and the manuscript has no major flaws, and all conclusions are well supported by the data.
2. The authors note that iRFP measurements were not observed to substantially increase, “probably due to nonlinear effects on fluorescence intensity at high concentrations of iRFP.” Indeed, the authors note in the discussion that PKR Δ -dependent improvements in “suppression” efficiency may be due to improvements in general translational initiation (i.e. “fixing” a GCE-related stress response). Why did the authors not further investigate this effect? The distinction between “GCE improvement” and “translational improvement” seems semantically important to make.
3. It is commonly noted in the literature that tRNA is often a limiting GCE component in mammalian expression contexts, and so it would reason that the typical solution of increasing tRNA copy number

(and therefore cellular concentrations) would contribute greatly to this RNA increase. Despite this, it is observed in Figures 1 and 3 that most interventions seem to have a more pronounced effect when tRNA levels are low. Could the authors comment on why these interventions seem to have a greater impact when tRNA levels are lowest, despite their putative effect purportedly intending to manifest at higher RNA levels? Does GCE actually increase RNA levels enough to cause PKR-dependent stress response? Or does GCE have some other impact that leads to these stress responses?

4. In many figures containing the FC plots of the iRFP-TAG-GFP reporter (i.e. Figure 1E), it can be seen that cells that have a greater GFP fluorescence also seem to have a greater iRFP fluorescence (i.e. the plot forms a “diagonal” pattern), indicating that greater GFP suppression seems to be correlated with greater iRFP fluorescence. This is in contrast to the graphical hypothesis depicted in Figure 1C. Could the authors comment on the phenomenon that gives rise to this effect? Does GFP FRET with iRFP? Or is this some form of artifact arising from nonlinearity in the iRFP fluorescent channel?

Minor Comments

1. Is the data presented in Figure 1D the same as is presented in the supplemental? If so, the caption should make this clear.
2. For figure 3, why did the authors only include a single replicate for eIF2 α treatments, while the PKR Δ treatments were performed in triplicate? Could the authors comment on this?
3. Why, in Figure 3E, does the “-ncAA” have higher levels of iRFP fluorescence than any of the other conditions? Could the authors comment on why adding ncAA or any of the other treatments have lower iRFP levels?

Reviewer #2 (Remarks to the Author):

Genetic code expansion (GCE) is a technique wherein orthogonal aminoacyl-tRNA synthetase and tRNA pairs are used to install noncanonical amino acids (ncAAs) into proteins in living cells. A major drawback of this technique is the low efficiency with which ncAAs are incorporated into proteins. Many strategies have been explored for improving the efficiency of ncAA incorporation. The manuscript by Lemke et al. describes the authors' efforts to improve the efficiency with which ncAAs are incorporated into proteins in mammalian cells by engineering the cellular stress response pathway. The stress response pathway has been shown to profoundly impact the rate of protein synthesis, and the authors hypothesize that the cell's response to stress conditions contributes to the low efficiency of ncAA incorporation. Three approaches are described to improve the efficiency of ncAA incorporation. All three approaches involve overexpressing proteins (i.e., the catalytic subunits of eIF2B, the N-terminal 174 residues of protein kinase R (PKR), and an eIF2 α (S51A) mutant) involved in the PKR-dependent eIF2 α phosphorylation pathway. The authors present data to support the conclusion that overexpression of these proteins can

improve the efficiency of ncAA incorporation using traditional GCE systems and, to a lesser extent, using film-like organelle GCE systems developed in the authors' laboratory.

Engineering the cellular stress response pathway is an appropriate and interesting strategy to improve the efficiency of GCE applications. It is noted that prior reports showed that overexpression of certain proteins in this stress response pathway (e.g., the N-terminal fragment of PKR) can lead to improvements in protein production. This reduces the novelty of the report somewhat. The manuscript would be of interest to GCE researchers. There are concerns involving experimental design and interpretation that should be addressed to support the stated conclusions.

Concerns:

1. A concern is the lack of transparent quantitative data. The manuscript provides only heatmaps to support the idea that engineering the cellular stress response pathway improves GCE. The claim is that, under certain conditions, the efficiency of GCE is improved by “~2.2 fold”; however, no quantitative data (or statistical analyses) are provided. For improved transparency and rigor, exact quantitative data (including standard deviation and data points) should be provided. Also, why use “~” when exact values can and should be reported?

2. Only one assay (GFP expression as measured by flow cytometry) is used to quantify the efficiency of GCE. Ideally, a second and complimentary assay would be used to confirm the data from the flow cytometry experiment.

3. The manuscript should address whether the increase in GFP production is a result of better ncAA incorporation. It is possible that the increase in GFP production is a result of an increase in stop-codon-readthrough under poorly regulated protein synthesis conditions. An analysis of the protein (i.e., mass spectrometry) to demonstrate that the ncAA is the only amino acid being incorporated under the optimized expression conditions would rule out this possibility.

4. The Integrated stress response, e.g. PKR and eIF2 phosphorylation, is central for the strategy of enhanced efficiency of GCE. Measures of the ISR should be included in the GCE assay. Does the engineered modifications of the ISR have the expected outcomes in the pathway markers?

Reviewer #3 (Remarks to the Author):

Summary: Sushkin et al. describe a method to improve production of non-canonical amino acid (ncAA) containing proteins in mammalian cells by modulating translation initiation through the protein kinase R (PKR) pathway. They test three strategies to engineer this pathway and find that the expression of a N-terminal fragment of PKR (PKRΔ) improves expression of GFP containing a premature TAG codon. Next, they evaluate this strategy to improve protein expression in orthogonally-translating organelles that are localized to different parts of the cell, including the plasma membrane, the Golgi membrane, the endoplasmic reticulum, and the outer mitochondrial membrane. They find that, regardless of expression of PKRΔ in the cytoplasm or in the OT organelle, the incorporation of ncAAs is maintained or slightly improved as a result. The localization appears to slightly affect ncAA incorporation, although the effect of the PKRΔ is relatively consistent. In total, they find that PKRΔ expression is a viable strategy for improving ncAA-incorporation in proteins.

Impact: The ability to spatially localize genetic code expansion to orthogonally-translation organelles is a creative way to prevent toxicity due to readthrough of endogenous TAG stop codons in the genome. Because this method is lower yielding compared to traditional overexpression protocols, this work by Sushkin et al. could be useful towards improving yields of proteins containing ncAAs in mammalian cells. However, certain claims about general improvements in protein yields are difficult to interpret based on the presented data. In addition, their data would benefit from further characterization to support improved ncAA incorporation due to higher protein expression levels. These features lowered my enthusiasm for the manuscript. Thus, as presented, the work feels a bit better suited for a more specialized journal.

Major concerns:

- At the end of the first results paragraph, it is stated that “Changes in the iRFP signal were also analyzed for all tested factors....probably due to nonlinear effects on fluorescence intensity at high concentrations of iRFP.” It would be difficult to believe the interpretation of the data presented in Figures S1A/B if the instrumentation is not providing an accurate measurement of iRFP levels, as “efficiency” in Fig S1A/B is defined as the ratio between GFP/iRFP levels. Comments about general protein expression levels are therefore difficult to believe, as RFP appears to be intended as a control for protein expression. Perhaps it would be possible to repeat only RFP expression in the presence and absence of the stress conditions they tested, but use an expression cassette that is weaker to provide more accurately quantify the expression levels.

- Due to the presumed improvements in translation occurring because of the introduction PKRΔ, it is necessary to assess whether the fidelity of translation is affected. Control experiments examining GFP and RFP fluorescence in the presence of the PKRΔ but in the absence of the ncAA should be conducted to measure the amount of background suppression of the TAG codon, both in the cytoplasm and in the orthogonally translating organelle. It may also be interesting to assess background readthrough of the

TAG codon in the presence of the EIF2B γ + EIF2B ϵ and the EIF2 α S15A as well. In addition, proteins should be analyzed by mass spectrometry to determine if ncAAs are still being incorporated efficiently.

Minor concerns:

- The reporter plasmid architecture for cytoplasmic expression should be clarified. Is this a fusion protein? Or is this a single plasmid with two expression cassettes? Related to the above question – if this is not a fusion protein, how would GFP expression affect iRFP expression? It appears in Figure 1E (-ncAA vs. Mock) that GFP expression reduces iRFP expression, for example. I think it would be useful to report both absolute values for iRFP and GFP signals, potentially as histograms, so that the magnitude of each product can be seen. This should apply to all flow data.

- “GCE efficiency” and “relative efficiency” is often used but not defined clearly. For example, in Figure 1D, relative efficiency is the ratio between median GFP signals; in Supplementary Figures 1A and 1B, relative efficiency is defined as the ratio between GFP/iRFP signals; and in Figure 3, relative efficiency is defined as the ratio between mCherry signals. This is confusing and makes interpretation of the data very difficult. Either different terms should be used to describe each of these, or they should be described explicitly (i.e. GFP_{stress} condition x/GFP_{mock} for Figure 1).

- Some reasoning on why they investigated the effect of modulating the stress response on the tRNA and not the aaRS mRNA / reporter mRNA would be appreciated.

- “Test with OT film-like organelles....GCE efficiency.” It is unclear what is meant by GCE efficiency in this sentence. Relative efficiency (in this case mCherry OT / mCherry cytoplasm) or fold change selectivity?

- The difference between Figure 3 and Fig S5 is very unclear – are these the same datasets but plotting median (Fig 3) and mean (Fig S5)? It is unclear why that would be necessary, as the difference between the two figures are not explained in the main text and the figure captions are nearly identical.

General response to all three reviewers

We thank all three reviewers for their thorough analysis of our paper and for insightful comments, that significantly helped to improve the paper. As detailed below, we added new experiments that address reviewers' comments, explain previously reported effects and further validate the strategy of the remodeling the cellular stress response for genetic code expansion (GCE) enhancement.

In our study, various strategies with different underlying mechanisms were compared side-by-side to tailor the protein kinase R-dependent eIF2 α phosphorylation pathway for higher GCE enhancement (please see new Fig. 1D,E and new Supplementary Figs. 1-4) . We now also additionally tested if a combination of best strategies could be applied to gain an even higher increase in GCE efficiency and showed that indeed such an approach enables more efficient GCE performance (new Fig. 1D,E and new Supplementary Figs. 9-10) .

In the new version of the manuscript, we addressed the question how increase in amber suppression efficiency correlates with the increase in overall protein expression and studied relations between these two parameters in more details. We now included new experiments where we tested previously used iRFP-GFP^{39TAG} reporter at different amounts (new Supplementary Figs. 11 and 12) and also provided new data where the new reporter GFP-P2A-T2A-mCherry^{189TAG} was applied to measure changes in the amber suppression efficiency and in the overall protein expression (new Supplementary Figs. 13-15). We showed that in some conditions fold increase in the amber suppression efficiency is higher than fold increase in the overall protein expression and in other conditions the opposite trend can be observed, thus highlighting the need for the careful estimations of these two parameters. However, the stress remodelers from different strategies exhibited the beneficial effect for GCE efficiency in all tested reporter systems, further validating the advantageous effect of the stress remodelers on the GCE enhancement.

We also performed new experiments, where the impact of remodeling the stress pathway on GCE realization was examined in other systems to validate the beneficial effect of stress remodelers' addition. Reporters with multiple (two or three) stop codons were investigated in the presence of various stress remodelers and an increase in GCE efficiency up to 3.2-fold was observed using cytoplasmic PyIRS (new Supplementary Figs. 17-21) or OT organelle (new Supplementary Figs. 41-43). Systems, where numerous stop codons are being reprogrammed for multiple noncanonical amino acids (ncAAs) incorporation into the protein of interest, are in high demand in versatile applications (e.g. FRET studies, antibody multilabeling).

Moreover, in our revision we now include new data showing the potential of using the PKR Δ stress remodeler for biotechnological applications, namely the expression of Trastuzumab (antibody binding to the HER2 receptor) for antibody-drug conjugates (ADCs) development (please see new Fig. 2), a market where two-fold yield increases could translate into substantial lower production cost. For many biotechnological applications, the yield of the actual final protein is really the key parameter, while the host physiology is not relevant as long as the host produces sufficient protein. We hope, that the reviewers agree, that this

extension showcases our research progress better for those cases where yield is the key optimisation criterion.

In the new version of the manuscript, we also investigated the potential problem of disturbed and inaccurate readthrough of the stop codon while using stress remodelers to enhance GCE. New experiments demonstrated the accuracy of the ncAA incorporation using flow cytometry (new Supplementary Figs. 10, 15, 19, 21, 25, 33, 43) and mass spectrometry analysis (new Supplementary Fig. 22).

We also further analyzed how PKR-based stress remodelers enhance OT organelle-associated GCE performance and added new experiments where PKR Δ anchored to the plasma membrane film-like compartment was tested to improve GCE efficiency for differently localized (at Golgi membrane, at ER membrane and at outer mitochondrial membrane) OT organelles. We could observe that such an anchored version of the PKR Δ could improve GCE efficiency for various OT organelles (new Supplementary Figs. 36-40) and this result illustrates the interesting interplay between organelles' microenvironment and the cytoplasm.

Last but not least, we expanded statistical analysis of our work, by doing triplicates for almost all experiments and added standard deviations with data points to bar plots where three biological experiments were performed (new Supplementary Figs. 1, 17, 23, 26, 27, 34, 35).

We hope that the reviewers can positively evaluate our research progress and improvements we added to the new version of the manuscript. We would like to thank the reviewers for their time and effort spent for assessing our manuscript.

Point-by-point

Reviewer comments in black

Our response in blue

Reviewer 1:

Sushkin et al., investigated the effects of intervening in the regulation of the protein kinase R (PKR)-dependent eIF2 α phosphorylation pathway, which according to the authors, is a stress response that occurs in mammalian cells during presence of heightened RNA levels (among other cellular stressors) such as those that may arise during genetic code expansion (GCE), which lowers translational initiation. To this effect, the authors explored 3 possible strategies to perturbing this pathway to the benefit of GCE: 1) introducing a binary eIF2B $\gamma\epsilon$ domain to catalyze GTP replacement in the eIF2 complex and thereby increase translation initiation; 2) introducing a truncated form of PKR (PKR Δ) that limits eIF2 inhibition; and 3) introducing a mutant form of eIF2 α (eIF2 α -S51A) that is resistant to phosphorylation (and therefore inhibition). Using an iRFP-TAG-GFP reporter system, the authors observed that the best intervention improved suppression efficiency and GFP fluorescence by up to 2.2-fold. The PKR Δ (at lower tRNA levels) having the most substantial effect, followed by eIF2 α -S51A. The authors then employ these two strategies to improve GCE efficiency on orthogonal, phase transitioned organelles. They observed that PKR Δ improved efficiency within organelles to a level approaching those of where GCE components were not targeted to orthogonal organelles, while improving selectivity. Interestingly, targeting the PKR Δ protein to these orthogonal organelles did not as markedly improve suppression efficiency as much as cytoplasmic PKR Δ . While targeting the stress response is an innovative approach to improve GCE efficiency the level of improvement was minimal.” “Targeting phase transition organelles is interesting but it is unclear from this application how this will advance protein engineering or cell biology. The minimal improvement in GCE demonstrated here will likely have a modest impact on the study of cellular function.

We thank the reviewer for the assessment of novelty of our concept for increasing genetic code expansion (GCE) efficiency. We respectfully disagree with the minimal improvement in GCE. Yield of GCE remains one of the major limitations of the method in cell biological applications, but in particular in biotechnological applications. Most “standard” amber suppression system yields already 20 to 40% compared with expression of the WT protein under standard conditions, and thus order of magnitude changes is not to be expected at the first place. How substantial an expected yield increase is, is certainly also subjective, as “reviewer 2” comments on our yield enhancement “as profound”.

We would like to persuade the reviewer by arguing that many developments in the GCE field that aim to use distinct strategies to enhance GCE, report a 2-4-fold increase in yield as a significant improvement, as we did in the present study –

e.g.

- Fan C. et al., Nucleic Acids Res., 2015 (rationally designed modified tRNA^{Pyl} provided 1.7-3-fold increase for the GFP signal using noncanonical amino acid Bock in *E.coli*),

- Serfling R. et al., Nucleic Acids Research, 2018 (newly engineered tRNAs enabled up to 3-fold increase the GFP signal using GFP reporter with two amber stop codons, Bock and HEK293T cells),
- Bryson D. et al., Nat Chem Biol., 2017 (evolved PylRS by phage-assisted continuous evolution method demonstrated up to 3.9-fold increase for the GFP signal using GFP reporter with three amber stop codons, Bock and *E.coli*),
- Schmied W. et al., JACS, 2014 (addition of mutated version eRF1 E55D leads best to 2.4-fold increase in GCE efficiency; GFP reporter with three stop codons, Bock and HEK293T cells were used).

Additionally, in the new paper version we included experiments where we showed that the combination of 2 strategies (PKRΔ + eIF2α S51A v2 stress remodeler) can further increase the yield of protein of interest (POI), achieving 2.8-fold increase in GCE efficiency in case of using the iRFP-GFP^{39TAG} reporter (new Fig. 1D,E and new Supplementary Figs. 1,9) or 3.4-fold increase for GFP and 3.2-fold increase for mCherry using GFP^{39TAG}, mCherry^{189TAG}::ms2 reporter respectively (new Supplementary Figs. 23-25).

We now also performed new experiments to show that our system helps to increase yield of POI, where two or three amber stop codons have been reprogrammed for ncAA incorporation (new Supplementary Figs. 16-21, 41-43). The new results show that PKRΔ + eIF2α S51A v2 stress remodeler allowed up to 3.2-fold increase in GFP signal using cytoplasmic PylRS and reporter iRFP-GFP^{39TAG,149TAG} (new Supplementary Figs. 16-19). OT organelles enabled also up to 3.2-fold increase in GCE efficiency using the reporter GFP^{39TAG,149TAG,182TAG}::ms2 (new Supplementary Figs. 41-43). Such results are particularly important for diverse experiments where incorporation of multiple ncAAs is highly recommended (e.g. FRET studies or antibody multilabeling).

With respect to the comment that “...Targeting phase transition organelles is interesting but it is unclear from this application how this will advance protein engineering or cell biology.” we have rewritten this section for clarity and also performed new experiments. In summary, our orthogonally translating (OT) organelles provide an opportunity for mRNA-selective GCE in mammalian cells and lead to decrease of unspecific ncAA incorporation into housekeeping proteins with a natural amber stop codon. Experiments, that are highly sensitive to background signal (e.g., highly resolving fluorescence imaging) could substantially benefit from the usage of OT systems. At the same time, OT systems are typically less efficient (in terms of yield) compared with the “standard” (cytoplasmic systems) for GCE realization and thus new ways for enhancement of OT-performed GCE will be crucial to explore.

Major comments:

1. The authors have developed a tool that improves the efficiency of GCE marginally, and nearly restores the efficiency deficit encountered through spatially orthogonalizing GCE-dependent translation to phase-transitioned organelles. These two experiments, lack the transformative impact to advance the field. Nevertheless, the quality of science is sound and the manuscript has no major flaws, and all conclusions are well supported by the data.

We are grateful for positive evaluation of our study. We have commented on the “marginally” above and hope that we persuaded the reviewer to adapt to the view of reviewer 2 on this issue, because a twofold change is substantial, and state of the art. We hope that the new data on Trastuzumab expression makes this more obvious. We would like to point out, that this is the first time the effect of remodeling the stress pathway on GCE is investigated. Even more so, we devoted a part of the work to the investigation of principles of GCE regulation in membraneless organelles using our previously published OT systems. OT organelles are new and provide a fundamentally new way for tailoring protein engineering. Logically, they are still understudied and their full potential needs to be explored. As stated by the reviewer, improvements of organelle-associated GCE performance up to the level of cytoplasmic PylRS, that we achieved for ERMP organelle (please see new Supplementary Fig. 27), demonstrates the possibility to realize GCE in mammalian cells at the same efficiency level as for common GCE system. However, the advantage is that using an OT organelle, this additionally happens in a protein-specific manner with much less background incorporation, which is highly advantageous for in-cell protein engineering. We are convinced that exploring the stress pathway in context of GCE engineering in standard cytoplasmic as well as in membraneless systems will inspire the community to use new ways how to engineer proteins in cells.

2. The authors note that iRFP measurements were not observed to substantially increase, “probably due to nonlinear effects on fluorescence intensity at high concentrations of iRFP.” Indeed, the authors note in the discussion that PKRΔ-dependent improvements in “suppression” efficiency may be due to improvements in general translational initiation (i.e. “fixing” a GCE-related stress response). Why did the authors not further investigate this effect? The distinction between “GCE improvement” and “translational improvement” seems semantically important to make.

It is absolutely true that tailoring of the cellular stress response itself is not necessarily mRNA-selective and can also influence translation rates for other proteins. Prompted by the reviewers’ request, we have expanded our analysis in this regard. New Supplementary Fig. 1 shows that there is, as we indicated before, a correlation between iRFP expression and GFP(TAG) expression. Notably, the beneficial effect of tailoring the stress response is in average higher for GCE then for normal translation of the iRFP reporter (e.g. 2.8-fold maximum increase vs 1.3-fold maximum increase in presence of PKRΔ + eIF2α S51A v2 stress remodeler using iRFP-GFP^{39TAG} reporter) when maximal plasmid mass for the reporter (300 ng) was used to simulate biotechnological conditions where the higher protein yield is always better. Remarkably, when iRFP-GFP^{39TAG} reporter was titrated down, transfecting 50 ng or 10 of the reporter, the higher effect for GFP increase over the iRFP increase was diminishing (new Supplementary Figs. 11 and 12).

We additionally used another fluorescent reporter GFP-P2A-T2A-mCherry^{189TAG} to assess GCE enhancement level in comparison with increase in general translation rate. We replaced iRFP with mCherry since good sites for amber suppression in mCherry are well-known, while for iRFP are not. Notably, mCherry(TAG) 1.6-fold increase was lower than 1.9-fold increase in GFP in case of PKRΔ + eIF2α S51A v2 addition (new Supplementary Figs. 13-15). Together, in this paper we for the first time demonstrate applicability of remodeling the stress pathway for

GCE enhancement to get higher yield of ncAA-modified POI and we now explain this in more details in the text.

3. It is commonly noted in the literature that tRNA is often a limiting GCE component in mammalian expression contexts, and so it would reason that the typical solution of increasing tRNA copy number (and therefore cellular concentrations) would contribute greatly to this RNA increase. Despite this, it is observed in Figures 1 and 3 that most interventions seem to have a more pronounced effect when tRNA levels are low. Could the authors comment on why these interventions seem to have a greater impact when tRNA levels are lowest, despite their putative effect purportedly intending to manifest at higher RNA levels? Does GCE actually increase RNA levels enough to cause PKR-dependent stress response? Or does GCE have some other impact that leads to these stress responses?

We corrected our workflow for flow cytometry data analysis to adjust gating for live cells to select accurately the whole main population of live cells (please see new Supplementary Fig. 44) and observed similar fold increase in GFP efficiency for all three tRNA^{Pyl} levels (2-fold for 300 ng tRNA^{Pyl}, 2.3-fold for 50 ng tRNA^{Pyl}, 2.2-fold for 10 ng tRNA^{Pyl} plasmid transfected, please see updated Fig. 1D and new Supplementary Fig. 1).

In general, when protein expression is already near optimum, as when using good reporters as GFP, also other effects can become limiting (e.g. total protein that can be produced within a given time frame) and thus dampen the effect of improvement. This is why titrating tRNA amounts is a good method to scan robustness of the affect across a larger dynamic range. By using Trastuzumab, we now add new data that shows the benefit is also significant for a “real” protein of wide interest (new Fig. 2).

4. In many figures containing the FC plots of the iRFP-TAG-GFP reporter (i.e. Figure 1E), it can be seen that cells that have a greater GFP fluorescence also seem to have a greater iRFP fluorescence (i.e the plot forms a “diagonal” pattern), indicating that greater GFP suppression seems to be correlated with greater iRFP fluorescence. This is in contrast to the graphical hypothesis depicted in Figure 1C. Could the authors comment on the phenomenon that gives rise to this effect? Does GFP FRET with iRFP? Or is this some form of artifact arising from nonlinearity in the iRFP fluorescent channel?

As discussed above, we have now investigated much more the correlation between iRFP and GFP(39TAG). The “diagonal” pattern indeed reflects an increase in both signals. In new Fig. 1C we show the general scheme regarding the rational design of the reporter. Two situations are demonstrated: in first case ncAA is not added and GFP signal cannot be observed; in second case ncAA is added, but still no stress remodeler is applied, such situation leads to the appearance of the GFP signal.

The spectral overlap between GFP and iRFP is very small (please see the Point-by-point Figure 1 below) and thus we do not expect to have efficient FRET in our experiment, and also did not notice this as a complication.

Point-by-point Figure 1. Excitation and emission spectra of GFP and iRFP.

Minor comments:

1. Is the data presented in Figure 1D the same as is presented in the supplemental? If so, the caption should make this clear.

For stress remodelers eIF2B γ + eIF2B ϵ , PKR Δ , eIF2 α S51A and PKR Δ + eIF2 α S51A v2 data is the same in new Fig. 1D and new Supplementary Fig. 1. We made this clearer in the caption to new Supplementary Fig. 1.

2. For figure 3, why did the authors only include a single replicate for eIF2 α treatments, while the PKR Δ treatments were performed in triplicate? Could the authors comment on this?

We briefly screened if there is any effect on organelle-performed GCE efficiency after addition of eIF2 α S51A and used 4 organelles for this test. We did not get a big increase in GCE efficiency for neither of tested organelles and thus only investigated those further that showed more promising results. Admittedly, these experiments are a lot of work, in particular when one does proper independent replicates and not just technical replicates, and this is why we did not perform triplicates when we did not detect substantial improvements. However, we agree with the reviewer that even for less efficient systems statistics should be thoroughly analyzed and thus now three independent experiments were added for eIF2 α S51A tests with OT organelles (new Supplementary Figs. 34 and 35).

3. Why, in Figure 3E, does the “-ncAA” have higher levels of iRFP fluorescence than any of the other conditions? Could the authors comment on why adding ncAA or any of the other treatments have lower iRFP levels?

We assume the reviewer in this comment means Fig. 1E, where a difference in color for a mild iRFP-positive population could be observed between “-ncAA” sample and other samples (“blue” density for “-ncAA” sample and “green” density for other samples). We note that for all samples we collected the same number of live cells. The reason why different color coding appeared is due to the different distribution of cells in the sample. For example, in the “-ncAA” sample there is no GFP-positive population and therefore a completely different distribution of cells takes place in the whole sample. Pseudocolor plots are built upon normalization within the one sample and so they help to compare cell densities within the one sample. Such pseudocolor plots can give an approximate estimation of changes in cell densities between 2 samples where cell distribution is similar, however they cannot be used

for comparison of cell numbers when distributions are different. To compare cell numbers between different samples cell gating should be applied as a precise and reliable approach. Please find below the Point-by-point Figure 2 where we show that indeed the same number of cells was located within the mild iRFP-positive gate.

Point-by-point Figure 2. FC plots showing number of cells within the mild iRFP-positive gate for samples with different transfected amount of tRNA^{Pyl} and tested in the absence or presence of SCO-K. Numbers in the plots mean percentage of cells lying in the highlighted black box.

Reviewer 2:

Genetic code expansion (GCE) is a technique wherein orthogonal aminoacyl-tRNA synthetase and tRNA pairs are used to install noncanonical amino acids (ncAAs) into proteins in living cells. A major drawback of this technique is the low efficiency with which ncAAs are incorporated into proteins. Many strategies have been explored for improving the efficiency of ncAA incorporation. The manuscript by Lemke et al. describes the authors' efforts to improve the efficiency with which ncAAs are incorporated into proteins in mammalian cells by engineering the cellular stress response pathway. The stress response pathway has been shown to profoundly impact the rate of protein synthesis, and the authors hypothesize that the cell's response to stress conditions contributes to the low efficiency of ncAA incorporation. Three approaches are described to improve the efficiency of ncAA incorporation. All three approaches involve overexpressing proteins (i.e., the catalytic subunits of eIF2B, the N-terminal 174 residues of protein kinase R (PKR), and an eIF2 α (S51A) mutant) involved in the PKR-dependent eIF2 α phosphorylation pathway. The authors present data to support the conclusion that overexpression of these proteins can improve the efficiency of ncAA incorporation using traditional GCE systems and, to a lesser extent, using film-like organelle GCE systems developed in the authors' laboratory. Engineering the cellular stress response pathway is an appropriate and interesting strategy to improve the efficiency of GCE applications.

We thank the reviewer for positive evaluation of our new concept for GCE enhancement.

It is noted that prior reports showed that overexpression of certain proteins in this stress response pathway (e.g., the N-terminal fragment of PKR) can lead to improvements in protein production. This reduces the novelty of the report somewhat. The manuscript would be of interest to GCE researchers.

We would like to point out that one large focus group of this paper are protein engineers, since we compare side-by-side different strategies and show the way for their further optimization (e.g. best approach to fuse remodelers from two different strategies, please see new Fig. 1D,E and new Supplementary Figs. 1, 9). However, we also address user-designed processes via the regulation in custom-designed membraneless organelles and thus our study might be useful in context of further development of novel synthetic organelles in a cell. This is a growing field, as it also makes functional use of concepts developed by the phase separation community. Ultimately, OT organelles are not just new membraneless organelles that execute a designed function, they also help us to study bottom-up how the unique biochemical environment of membraneless organelles in general might underlie principles of cellular function.

There are concerns involving experimental design and interpretation that should be addressed to support the stated conclusions.

Concerns:

1. A concern is the lack of transparent quantitative data. The manuscripts provides only heatmaps to support the idea that engineering the cellular stress response pathway improves GCE. The claim is that, under certain conditions, the efficiency of GCE is improved by “~2.2

fold”; however, no quantitative data (or statistical analyses) are provided. For improved transparency and rigor, exact quantitative data (including standard deviation and data points) should be provided. Also, why use “~” when exact values can and should be reported?

We apologize for this and following the reviewers’ comments we have substantially expanded our experimental work, did triplicates for almost all experiments and then reworked our statistical analysis. Admittedly, these experiments are a lot of work, in particular when one does proper independent replicates and not just technical replicates, and this is why we did not perform triplicates when we did not detect substantial improvements. However, we agree with the reviewer that even for less efficient systems statistics should be thoroughly analyzed. We now include bar plots with standard deviations (new Supplementary Figs. 1, 17, 23, 26, 27, 34, 35) to show statistical analysis of obtained data. We also provided exact numbers for efficiency fold improvement in the paper. We chose heat maps for compact visual representation of our data, feeling that this is an efficient way to deal with larger data sets. We also note that each flow cytometry plot is at least from 100000 live cells, and thus has very good statistics. Now we provide the raw flow data in the substantially expanded supporting information file to show how observed effects for GCE and general translation enhancement appear at FC plots.

2. Only one assay (GFP expression as measured by flow cytometry) is used to quantify the efficiency of GCE. Ideally, a second and complimentary assay would be used to confirm the data from the flow cytometry experiment.

To address the reviewer comment we have greatly expanded our work. With the new data we show significant increase in the yield of POI with two and three introduced amber stop codons for cytoplasmic PylRS and OT organelles, achieving up to 3.2-fold increase in GFP signal using cytoplasmic PylRS and reporter iRFP-GFP^{39TAG,149TAG} (new Supplementary Figs. 17-19) or using OT organelle and reporter GFP^{39TAG,149TAG,182TAG}::ms2 (new Supplementary Figs. 41-43). Moreover, we demonstrate the application of PKRΔ to enhance expression of ncAA-modified antibody Trastuzumab, which is not just a simple model reporter, but a protein which sparks high interest for bioconjugation to generate ADCs for therapeutic purposes (new Fig. 2). Despite a completely different expression system (secretion of Trastuzumab in medium out of mammalian cells), we obtained consistent results that tailoring the stress pathway can enhance expression of POI.

3. The manuscript should address whether the increase in GFP production is a result of better ncAA incorporation. It is possible that the increase in GFP production is a result of an increase in stop-codon-readthrough under poorly regulated protein synthesis conditions. An analysis of the protein (i.e., mass spectrometry) to demonstrate that the ncAA is the only amino acid being incorporated under the optimized expression conditions would rule out this possibility.

This is a very good point, and we agree with the reviewer and have now expanded on this substantially with new experiments. In the new version of the manuscript, we included additional controls, where samples were transfected with one of the applied reporters, GCE machinery and a stress remodeler, but were not incubated with ncAA (new Supplementary Fig. 10, 15, 19, 21, 25, 33, 43). For such controls we could observe no or small GFP(TAG) or

mCherry(TAG)-positive population only when high tRNA^{Pyl} levels were transfected. Additionally, this GFP(TAG) or mCherry(TAG)-positive population was getting smaller from samples with higher to lower amount of transfected tRNA^{Pyl}, implying a suppressor tRNA^{Pyl}-dependent decrease in intensity of GFP(TAG) or mCherry(TAG) population in the absence of ncAA.

We also did “minus ncAA, plus stress remodeler” controls in the new experiment where we tested effects of stress remodelers’ addition on production of ncAA-modified Trastuzumab and could not detect any signal with enzyme-linked immunosorbent assay for samples incubated with stress remodeler, but without ncAA (new Fig. 2). We additionally performed mass spectrometry analysis of ncAA-modified Trastuzumab and we detected incorporation of only the applied ncAA at the introduced site of stop codon validating our hypothesis. The new data is included in new Supplementary Fig. 22.

4. The Integrated stress response, e.g. PKR and eIF2 phosphorylation, is central for the strategy of enhanced efficiency of GCE. Measures of the ISR should be included in the GCE assay. Does the engineered modifications of the ISR have the expected outcomes in the pathway markers?

In this paper we focus on adding stress remodelers on top of the GCE machinery to boost ncAA-modified protein production and develop the best strategies for the GCE enhancement. Indeed, we assume that many changes at different levels can take place in the PKR-dependent eIF2 α phosphorylation pathway after addition of tested stress remodelers and because the ISR pathway is complex, we already tested three different strategies. We agree that it would be interesting to study the ISR pathway in the context of GCE and especially organelle-associated GCE, but we hope the reviewer can agree that these studies would be extensive to arrive at a satisfying conclusion and are thus beyond the scope of this paper.

Reviewer 3:

Summary: Sushkin et al. describe a method to improve production of non-canonical amino acid (ncAA) containing proteins in mammalian cells by modulating translation initiation through the protein kinase R (PKR) pathway. They test three strategies to engineer this pathway and find that the expression of a N-terminal fragment of PKR (PKR Δ) improves expression of GFP containing a premature TAG codon. Next, they evaluate this strategy to improve protein expression in orthogonally-translating organelles that are localized to different parts of the cell, including the plasma membrane, the Golgi membrane, the endoplasmic reticulum, and the outer mitochondrial membrane. They find that, regardless of expression of PKR Δ in the cytoplasm or in the OT organelle, the incorporation of ncAAs is maintained or slightly improved as a result. The localization appears to slightly affect ncAA incorporation, although the effect of the PKR Δ is relatively consistent. In total, they find that PKR Δ expression is a viable strategy for improving ncAA-incorporation in proteins.

Impact: The ability to spatially localize genetic code expansion to orthogonally-translation organelles is a creative way to prevent toxicity due to readthrough of endogenous TAG stop codons in the genome. Because this method is lower yielding compared to traditional overexpression protocols, this work by Sushkin et al. could be useful towards improving yields of proteins containing ncAAs in mammalian cells.

We thank the reviewer for positive remarks regarding our study and for acknowledging the relevance of yield enhancement for GCE in general and for OT based systems – which are lower yielding – in particular.

However, certain claims about general improvements in protein yields are difficult to interpret based on the presented data. In addition, their data would benefit from further characterization to support improved ncAA incorporation due to higher protein expression levels. These features lowered my enthusiasm for the manuscript. Thus, as presented, the work feels a bit better suited for a more specialized journal.

We have expanded our work to further characterize effects of stress remodelers in many more reporter systems: i) reporter bearing multiple amber sites (iRFP-GFP^{39TAG,149TAG} and iRFP-GFP^{39TAG,149TAG,182TAG}, please see new Supplementary Figs. 16-21), ii) reporter with another fluorescent protein design (GFP-P2A-T2A-mCherry^{189TAG}, new Supplementary Figs. 14 and 15), iii) titrated levels of iRFP-GFP^{39TAG} reporter (new Supplementary Figs. 11 and 12) and iv) we now also show the power of the concept for biotechnological purposes by providing new results on ncAA-modified Trastuzumab, which has great potential in antibody-drug conjugate development (new Fig. 2).

We also expanded our study on how addition and different localization of stress remodelers can enhance GCE realized by OT organelles, which are new and understudied compartments designed in mammalian cells. With these new experiments we hope we could better demonstrate the beneficial effect of stress remodelers' addition on the GCE efficiency and also show the applicability of stress remodeling concept for different GCE-related applications.

Major concerns:

- At the end of the first results paragraph, it is stated that “Changes in the iRFP signal were also analyzed for all tested factors....probably due to nonlinear effects on fluorescence intensity at high concentrations of iRFP.” It would be difficult to believe the interpretation of the data presented in Figures S1A/B if the instrumentation is not providing an accurate measurement of iRFP levels, as “efficiency” in Fig S1A/B is defined as the ratio between GFP/iRFP levels. Comments about general protein expression levels are therefore difficult to believe, as RFP appears to be intended as a control for protein expression. Perhaps it would be possible to repeat only RFP expression in the presence and absence of the stress conditions they tested, but use an expression cassette that is weaker to provide more accurately quantify the expression levels.

Prompted by this and the other reviewers request, we have expanded our work to further characterize effects of stress remodelers in multiple new reporter systems, using titrated levels of iRFP-GFP^{39TAG} reporter, reporter with another compilation of fluorescent proteins (GFP-P2A-T2A-mCherry^{189TAG}) or reporter bearing multiple amber sites (iRFP-GFP^{39TAG,149TAG} and iRFP-GFP^{39TAG,149TAG,182TAG}). New supplementary Fig. 1 shows that there is as expected a correlation between iRFP expression and GFP expression when using iRFP-GFP^{39TAG} reporter for testing the effect of addition of stress remodelers. Notably, the beneficial effect of tailoring the stress response is in average higher for GCE than for normal translation of the iRFP reporter (e.g. 2.8-fold maximum increase vs 1.3-fold maximum increase in presence of PKRΔ + eIF2α S51A v2 stress remodeler using iRFP-GFP^{39TAG} reporter) when maximal plasmid mass for the reporter (300 ng) was used. We also added the new experiments, where iRFP-GFP^{39TAG} reporter was titrated down (using 50 ng or 10 ng of the reporter plasmid) and we observed that the higher effect for GFP increase over the iRFP increase was diminishing (new Supplementary Figs. 11 and 12).

Next, we also designed a new reporter GFP-P2A-T2A-mCherry^{189TAG} to evaluate increase in GCE efficiency and in the translation boost after addition of various stress remodelers. When using 300 ng of the GFP-P2A-T2A-mCherry^{189TAG} reporter, we observed a lower or similar increase in GCE efficiency (mCherry expression for this reporter) compared with the fold increase in general translation (GFP expression for this reporter) after addition of stress remodelers, e.g. 1.6-fold increase in GCE efficiency and 1.9-fold increase in general translation after addition of stress remodeler PKRΔ + eIF2α S51A v2 (new Supplementary Figs. 13-15). Such results are in contrast to results obtained while using 300 ng iRFP-GFP^{39TAG} reporter, where addition of efficient stress remodelers led to higher fold increase in GCE efficiency compared with the fold increase in general translation. These new experiments highlight the need for the careful estimations of these two parameters when testing effects of stress remodelers' addition. However, introduction of stress remodelers enabled to enhance GCE in different reporter systems, validating the beneficial effect of addition of stress remodelers to improve GCE efficiency.

In the new version of the manuscript, we also added new experiments where we tested reporters bearing two (iRFP-GFP^{39TAG,149TAG}) or three stop codons (iRFP-GFP^{39TAG,149TAG,182TAG}) to assess increase in multiple ncAAs incorporation using cytoplasmic PylRS. Remodeling the cellular stress response allowed to achieve up to 3.2-fold in GCE efficiency after addition of stress remodeler PKRΔ + eIF2α S51A v2 using iRFP-GFP^{39TAG,149TAG} reporter (new Supplementary Figs. 17-21). Remarkably, application of OT organelle also provides up to 3.2-fold increase in multiple ncAAs incorporation when it was tested with GFP^{39TAG,149TAG,182TAG}::ms2 reporter (new Supplementary Figs. 41-43).

Additionally, we show the power of concept for biotechnological purposes by providing new results on ncAA-Trastuzumab, which has great potential in ADCs development. Here we report a 1.7-fold increase in ncAA-modified antibody, which would lower the cost of production accordingly (new Fig. 2).

- Due to the presumed improvements in translation occurring because of the introduction PKRΔ, it is necessary to assess whether the fidelity of translation is affected. Control experiments examining GFP and RFP fluorescence in the presence of the PKRΔ but in the absence of the ncAA should be conducted to measure the amount of background suppression of the TAG codon, both in the cytoplasm and in the orthogonally translating organelle. It may also be interesting to assess background readthrough of the TAG codon in the presence of the EIF2By + EIF2Be and the EIF2α S15A as well. In addition, proteins should be analyzed by mass spectrometry to determine if ncAAs are still being incorporated efficiently.

We agree that it is important to address the amount of background readthrough and thank the reviewer for drawing out attention to this. We now added experiments in which we transfected samples with one of the studied reporters, cytoplasmic PylRS or OT organelle, different amount of suppressor tRNA^{Pyl} (10, 50 or 300 ng) and a stress remodeler, but did not further incubate them with ncAA (new Supplementary Fig. 10, 15, 19, 21, 25, 33, 43). For such samples we could observe no or small GFP(TAG) or mCherry(TAG)-positive population only when high tRNA^{Pyl} levels were transfected. Remarkably, the intensity of this GFP(TAG) or mCherry(TAG)-positive population diminished as the amount of transfected tRNA^{Pyl} decreased. These results indicate that the changes in intensity of the GFP(TAG) or mCherry(TAG)-positive population in the absence of ncAA were suppressor tRNA^{Pyl}-dependent.

We performed the same type of controls (where stress remodeler was added to the samples while ncAA was not) in the new experiment where we test the increase in production of ncAA-modified antibody Trastuzumab after introduction of various stress remodelers. No signal was detected using enzyme-linked immunosorbent assay for samples incubated with stress remodeler, but without ncAA (new Fig. 2). Additionally, we carried out mass spectrometry analysis of purified ncAA-modified Trastuzumab, which showed incorporation of only applied ncAA (SCO-K) at the site of introduced stop codon, thus demonstrating accurate ncAA incorporation at the desired site of POI in the presence of applied stress remodeler PKRΔ (new Supplementary Fig. 22).

Minor concerns:

- The reporter plasmid architecture for cytoplasmic expression should be clarified. Is this a fusion protein? Or is this a single plasmid with two expression cassettes? Related to the above question – if this is not a fusion protein, how would GFP expression affect iRFP expression? It appears in Figure 1E (-ncAA vs. Mock) that GFP expression reduces iRFP expression, for example. I think it would be useful to report both absolute values for iRFP and GFP signals, potentially as histograms, so that the magnitude of each product can be seen. This should apply to all flow data.

The reporter is a fusion protein iRFP-linker-GFP^{39TAG}. For plasmid backbone and promoter information we provide now all information in Supplementary Table 1.

We agree with the point, that visually a difference in color for a mild iRFP-positive population could be observed between “-ncAA” sample and other samples (“blue” density for “-ncAA” sample and “green” density for other samples). We note that for all samples we collected the same number of live cells. The reason why different color coding appeared is due to the different distribution of cells in the sample. For example, in the “-ncAA” sample there is no GFP-positive population and therefore a completely different distribution of cells takes place in the whole sample. Pseudocolor plots are built upon normalization within the same sample and so they help to compare cell densities within the same sample. Such pseudocolor plots can give an approximate estimation of changes in cell densities between 2 samples where cell distribution is similar, however they cannot be used for comparison of cell numbers when distributions are different. To compare cell numbers between different samples cell gating should be applied as a precise and reliable approach. Please see Point-by-point Figure 2 (page 8) where we show that indeed the same number of cells was located within the mild iRFP-positive gate.

We also now provide additional Excel files with the submission, where all absolute numbers could be found and all calculated values for all performed experiments are listed. We also want to note that absolute values for different replicates in flow cytometry experiments can vary due to day-to-day changes in laser performance and therefore for calculations we use relative parameters, such as relative efficiencies. For each data point for relative efficiencies, we normalize the sample and the control that were measured at the same time 1 day after simultaneous and identical treatment. Such relative parameters provide more accurate results.

- “GCE efficiency” and “relative efficiency” is often used but not defined clearly. For example, in Figure 1D, relative efficiency is the ratio between median GFP signals; in Supplementary Figures 1A and 1B, relative efficiency is defined as the ratio between GFP/iRFP signals; and in Figure 3, relative efficiency is defined as the ratio between mCherry signals. This is confusing and makes interpretation of the data very difficult. Either different terms should be used to describe each of these, or they should be described explicitly (i.e. GFP_{stress} condition x/GFP_{mock} for Figure 1).

We thank the reviewer for this comment and we have improved the paper to make it clearer to understanding. We introduced new terms: relative efficiency^{GFP} (%); relative efficiency^{iRFP} (%); relative efficiency^{mCherry} (%) and describe them in the main text and captions to navigate readers which of the term we use in particular case to assess increase in signal of fluorescent protein.

- Some reasoning on why they investigated the effect of modulating the stress response on the tRNA and not the aaRS mRNA / reporter mRNA would be appreciated.

Suppressor tRNAs were selected for titration mainly because the amount of tRNA has been shown to be a limiting factor in GCE systems. In general, we note that our U6 tRNA expression plasmid for PylRS and optimized transfection conditions are simple and easy for transfection of HEK293T cells and have been optimized by us and the community for a long time and are near optimum of what the cell can produce for a simple reporter protein (like GFP) in those cells and thus we are near the saturation regime. Such conditions might not be representative for more “difficult expression” settings. Lower concentration of tRNA decreases overall GCE yield and thus can make it more likely to identify enhancing effects. In the new version of the paper, we also present results, where we titrated down iRFP-GFP^{39TAG} reporter (new Supplementary Figs. 11 and 12). In addition, we show, that the effect is robust for various double and triple amber constructs (new Supplementary Figs. 16-21), as well as for Trastuzumab (new Fig. 2).

- “Test with OT film-like organelles....GCE efficiency.” It is unclear what is meant by GCE efficiency in this sentence. Relative efficiency (in this case mCherry OT / mCherry cytoplasm) or fold change selectivity?

We were referring to relative efficiency^{mCherry} (%) in this sentence. We corrected this sentence to clarify which exactly parameter we used in that case to assess changes in GCE efficiency. In the new version of the paper, we also explained in the text and captions all parameters we calculated: relative efficiency^{GFP} (%), relative efficiency^{iRFP} (%), relative efficiency^{mCherry} (%) and fold change selectivity.

- The difference between Figure 3 and Fig S5 is very unclear – are these the same datasets but plotting median (Fig 3) and mean (Fig S5)? It is unclear why that would be necessary, as the difference between the two figures are not explained in the main text and the figure captions are nearly identical.

Old Figure 3 and old Fig S5 reported the same measurements, but two different ways of analysis. Flow cytometry plots (new Supplementary Figs. 29-32) were analyzed and for each plot mean and median values were exported. After that, median values were used for quantification of percentage relative efficiency^{mCherry} (normalized to each OT organelle without stress remodeler) and fold change selectivity of mCherry over GFP expression to obtain values for heat maps presented in new Fig. 3C,D. Mean values were used to calculate percentage relative efficiency^{mCherry} (normalized to NES-PylRS^{AF}) and fold change selectivity for heat maps presented in the new Supplementary Fig. 28.

The reason why 2 different approaches are presented, is that initially mean values were used for quantification of OT organelle performance in previous papers (Reinkemeier C.D. et al., *Science*, 2019; Reinkemeier C.D. & Lemke E.A., *Cell*, 2021, Reinkemeier C.D. & Lemke E.A., *J. Mol. Biol.*, 2022) however here we mostly used median values. Even though both approaches are applicable for data analysis and further parameter quantification, median values provide clearer view on a location of average cell in skewed populations. We additionally quantified parameters on base of mean values as in previous papers for Supplementary Fig. 28 to provide an overview of what level of GCE enhancement we could achieve to compare with previously published data for OT organelles.

REVIEWER COMMENTS

Reviewer #1 (Remarks to the Author):

Overall, the authors have responded adequately to all reviewer concerns. The authors have increased the examples, and rigor at which demonstrated how altering the stress response can increase the yield of proteins using genetic code expansion with amber codons. It still unclear if this is a TAG specific phenomena or enhancement in general protein expression and if the impact of this finding will alter studies using GCE or scaling protein production with GCE. The data does clearly support that the cellular stress response enhance GCE in mammalian cells.

Reviewer #2 (Remarks to the Author):

This is a revised manuscript that addresses strategies for improving the efficiency of nCAA incorporation by engineering the PKR/Integrated stress response (ISR). The prior concerns were largely addressed, with the exception of concern #4. This is an interesting manuscript that more rigorously develops ISR themes in the literature and applies them to expression of proteins with nCAA incorporation.

Concerns:

1. (Original concern # 4). The Integrated stress response, e.g. PKR and eIF2 phosphorylation, is central for the strategy of enhanced efficiency of GCE. Measures of the ISR should be included in the GCE assay. Does the engineered modifications of the ISR have the expected outcomes in the pathway markers?

The authors respond that they "assume that many changes at different levels can take place in the PKR-dependent eIF2 α phosphorylation pathway after addition of tested stress remodelers and because the ISR pathway is complex, we [they] tested three different strategies." Furthermore, the authors indicate that testing whether the ISR was modulated as proposed with one or more of the PKR/ISR modifying strategies is suggested to be beyond "the scope of the paper." Given that the measurements of endogenous ISR assays (e.g. a couple of measurements of mRNA or protein levels of key PKR/ISR biomarkers that are assumed to be interfered with) is straightforward, and this would experimental address one of the central conclusions of the manuscript, this reviewer would think that this would be an important experiment to include in the manuscript

2. It would be helpful to designate statistical significance in bar graphs, along with information about the analyses in the figure legends.

Reviewer #3 (Remarks to the Author):

The authors have done a nice job responding to reviewers, and have put a lot of data into this paper. I commend them for the effort. In terms of the new figure 2, can the authors show that they have a statistically significant improvement? I apologize if I missed it. Also, I think it would strengthen the paper and is important to create a new figure in the main body from the supplemental data showing multiple reporter systems, and improved protein expression. Please note I am not asking for new experiments, just re-arranging of the data.

Point-by-point

Reviewer comments in black

Our response in blue

Major changes in main text of the manuscript and in the Supplementary information are in yellow

Reviewer 1:

Overall, the authors have responded adequately to all reviewer concerns. The authors have increased the examples, and rigor at which demonstrated how altering the stress response can increase the yield of proteins using genetic code expansion with amber codons.

It still unclear if this is a TAG specific phenomena or enhancement in general protein expression and if the impact of this finding will alter studies using GCE or scaling protein production with GCE. The data does clearly support that the cellular stress response enhance GCE in mammalian cells.

We thank the reviewer for positive evaluation of our efforts to respond the comments of the reviewers and to improve the article with newly added experiments.

Reviewer 2:

This is a revised manuscript that addresses strategies for improving the efficiency of ncAA incorporation by engineering the PKR/Integrated stress response (ISR). The prior concerns were largely addressed, with the exception of concern #4. This is an interesting manuscript that more rigorously develops ISR themes in the literature and applies them to expression of proteins with ncAA incorporation.

We thank the reviewer for overall positive comment regarding our last submitted version of the manuscript.

Concerns:

1. (Original concern # 4). *The Integrated stress response, e.g. PKR and eIF2 phosphorylation, is central for the strategy of enhanced efficiency of GCE. Measures of the ISR should be included in the GCE assay. Does the engineered modifications of the ISR have the expected outcomes in the pathway markers?*

The authors respond that they "assume that many changes at different levels can take place in the PKR-dependent eIF2 α phosphorylation pathway after addition of tested stress remodelers and because the ISR pathway is complex, we [they] tested three different strategies." Furthermore, the authors indicate that testing whether the ISR was modulated as proposed with one or more of the PKR/ISR modifying strategies is suggested to be beyond "the

scope of the paper." Given that the measurements of endogenous ISR assays (e.g. a couple of measurements of mRNA or protein levels of key PKR/ISR biomarkers that are assumed to be interfered with) is straightforward, and this would experimental address one of the central conclusions of the manuscript, this reviewer would think that this would be an important experiment to include in the manuscript.

Since we use transient multi-plasmid transfection (we used 4 plasmid transfection and had ~30% reporter positive cells), which has its limits in terms of generating a homogeneous cell population. It is indeed not that straightforward to detect potentially mild global changes in ISR status in a heterogeneous cell culture. However, thanks to the reviewer's persistence and the added time the editor gave us, we have meanwhile optimized our protocols to perform the wanted experiments.

To assess how implementation of designed stress remodelers affects the ISR, we now performed two complimentary assays (Rabouw H. et al., PNAS, 2019; Cagnetta et al., Mol. Cell, 2019; Zyryanova A. et al., Mol. Cell, 2021): we investigated phosphorylation of eIF2 α and global protein synthesis rate using Western blots and added the new experiments to the new version of the manuscript as new Supplementary Figs. 16 and 17. The figures are also shown at the end of this point-by-point for convenience as Point-by-Point Figs. 1 and 2.

Western blot using anti-phospho-eIF2 α -antibody (Point-by-point Figs. 1, 2 and new Supplementary Figs. 16 and 17): We showed that after addition of PKR Δ the level of phosphorylated eIF2 α decreases, which is consistent with the previous studies and our results where addition of PKR Δ led to higher overall protein expression and to GCE enhancement (new Supplementary Fig. 16A,C). We also detected lower level of phosphorylated eIF2 α in case of eIF2B γ + eIF2B ϵ stress remodeler introduction (new Supplementary Figs. 16A,C), that potentially could be explained by higher formation of eIF2B complex which competes with kinases for interaction with eIF2 α and therefore higher amount of eIF2B complexes drags eIF2 complex out of the phosphorylation reaction towards the promotion of translation initiation. Unfortunately, reliable detection of reduction in eIF2 α phosphorylation for samples where eIF2 α S51A was overexpressed (stress remodelers eIF2 α S51A and PKR Δ + eIF2 α S51A v2) is not possible due to existing cross-talk between anti-phospho-eIF2 α antibody and non-phosphorylated eIF2 α (Point-by-point Fig. 2 and new Supplementary Fig. 17).

Global protein synthesis rate analysis (Point-by-point Fig. 1 and new Supplementary Fig. 16): Furthermore, we also inspected the global protein synthesis rate by evaluating a puromycin incorporation assay. Here, likely due to the mild effect due to transient transfection, which only incorporates puromycin into those few proteins made during the short expression period in those cells in which substantial amount of stress remodeler plasmid was transfected properly, changes are mild, but consistent and in trend with our expectations and with the anti-phospho-eIF2 α -antibody data. Gratifyingly, we can detect a statistically significant effect for our best condition (PKR Δ + eIF2 α S51A v2 stress remodeler), which was the most efficient one almost in all performed experiments (new Supplementary Figs. 16B,D) and also the one that we cannot probe with the anti-phospho-eIF2 α -antibody due to the cross-talk.

Together, all our new data support our rational assumption for usage of designed stress remodelers and provide an overview of the ISR status in GCE-active cells and after cellular stress remodeling.

2. It would be helpful to designate statistical significance in bar graphs, along with information about the analyses in the figure legends.

We thank the reviewer for the helpful comment and in new version of the manuscript we added results of statistical significance tests to the figures with bar plots (revised Fig. 3, new Supplementary Figs. 1, 16, 18, 24, 27, 28, 35, 36).

For ELISA measurements (revised Fig. 3) we measured three biological replicates and always had three dilutions for each sample to precisely quantify the antibody concentration in a sample and so we assume we have reliable statistics for the analysis, however, we now additionally expanded our statistical analysis on one-way ANOVA with Dunnett's multiple comparison test to show the significant difference between +/- PKRΔ samples.

For flow cytometry we like to point out that each flow cytometry (FC) plot reports on 100000 live cells, where target population even in samples with lowest signal (e.g. samples with 3xTAG reporter) is at least 1 % (and so 1000 live cells). Coefficient of variation (CV) can be calculated as $100 / \sqrt{\text{number of target cells}}$, giving value 3.16 % for 1000 cells, which is acceptable. For samples with higher signal CV value will be even lower, showing that we collect enough cells for quantification analysis. We even made the effort to do biological replicates, and not just technical replicates, and thus felt that the conservative plotting of the standard deviation is a very good way of assessing robustness. To properly calculate statistical significance in terms of p values, common tools like t-test / ANOVA were not our preferred method, since they are formally not defined for low number of samples. However, we do appreciate that it is still common to do this, and we now expanded our statistical analysis on t-testing and ANOVA to complement our data evaluation. The additional results of statistical significance tests were added to the figures with bar plots (revised Fig. 3, new Supplementary Figs. 1, 16, 18, 24, 27, 28, 35, 36) and discussed in more details in the method section.

Reviewer 3:

The authors have done a nice job responding to reviewers, and have put a lot of data into this paper. I commend them for the effort.

We thank the reviewer for the high evaluation of our efforts to respond the addressed questions and concerns of reviewers and to make our manuscript better with additional experiments.

In terms of the new figure 2, can the authors show that they have a statistically significant improvement? I apologize if I missed it.

We thank the reviewer for the helpful comment and in new version of the manuscript we added results of statistical significance tests to the figures with bar plots (revised Fig. 3, new Supplementary Figs. 1, 16, 18, 24, 27, 28, 35, 36).

For ELISA measurements (revised Fig. 3) we measured three biological replicates and always had three dilutions for each sample to precisely quantify the antibody concentration in a sample and so we assume we have reliable statistics for the analysis, however, we now additionally expanded our statistical analysis on one-way ANOVA with Dunnett's multiple comparison test to show the significant difference between +/- PKRΔ samples.

Also, I think it would strengthen the paper and is important to create a new figure in the main body from the supplemental data showing multiple reporter systems, and improved protein expression. Please note I am not asking for new experiments, just re-arranging of the data.

Following this comment, we relocated data for cytoplasmic GCE enhancement in systems with multiple ncAA incorporation from supplemental data to the main text (new Fig. 2), since we believe these experiments are relatively important and show an example of distinct reporter system where addition of stress remodelers can be beneficial for GCE efficiency. Experiments with multiple ncAA incorporation systems also have a big significance for practical applications like FRET studies or synthetic polymer synthesis in cells, since advantageous concepts from such experiments can potentially be transferred and used in aforementioned applications. We hope that we correctly understood the request from the reviewer and made it in an expected way.

Point-by-Point Figs.:

Point-by-point Fig. 1 (new Supplementary Fig. 16): **Assessment of the ISR status after addition of stress remodelers to GCE-performing cells.** (A, B) Representative Western blots for evaluation of phosphorylated and total eIF2α levels (A) and puromycin incorporation (B). Cyclophilin B was used as a loading control. Samples: 1) untransfected cells; 2) only Mock (1200 ng plasmid) transfected; 3) GCE (PylRS, tRNA^{Pyl}) with mock plasmid (no stress remodeler) supplemented without SCO-K; 4) GCE with mock plasmid (no stress remodeler) supplemented with SCO-K; 5) GCE with eIF2By + eIF2Be stress remodeler plus SCO-K; 6) GCE with PKRΔ stress remodeler plus; 7) GCE with eIF2α S51A stress remodeler plus SCO-K; 8) GCE with PKRΔ + eIF2α S51A v2 stress remodeler plus SCO-K; 9) untransfected cells under stress conditions (0.5 mM sodium arsenite, 1 hour incubation); 10) untransfected cells supplemented with distilled water instead of sodium arsenite; 11) untransfected

cells supplemented with water instead of puromycin. **(C, D)** Bar plots representing normalized ratios calculated after Western blot quantification for assessment of phosphorylated eIF2 α level **(C)** and puromycin incorporation **(D)**. Normalized ratio (P-eIF2 α) or (puromycin) was defined as P-eIF2 α **(C)** or puromycin **(D)** signal divided by cyclophilin B signal for each sample and normalized to the control sample - GCE with no stress remodeler addition (Mock) supplemented with SCO-K (sample 4 on the Western blots). Bar plots show the mean value for normalized ratios of five independent experiments, error bars represent the SD. Ns denotes not significant (p value > 0.05), * - p value ≤ 0.05 , ** - p value ≤ 0.01 , p values were calculated using one-sample (two-tailed) t -test.

Point-by-point Fig. 2 (new Supplementary Fig. 17): **Assessment of the ISR status in HEK293T cells transfected with eIF2 α S51A-containing stress remodelers and without GCE machinery (PyIRS and tRNA^{Pyl}).** Western blot demonstrates levels of phosphorylated and total eIF2 α in tested samples. Cyclophilin B was used as a loading control. Samples: 1) GCE with eIF2 α S51A stress remodeler plus SCO-K; 2) GCE with PKR Δ + eIF2 α S51A v2 stress remodeler plus SCO-K; 3) untransfected cells under stress conditions (0.5 mM sodium arsenite, 1 hour incubation); 4) untransfected cells supplemented with distilled water instead of sodium arsenite.

REVIEWERS' COMMENTS

Reviewer #2 (Remarks to the Author):

This is an R2 manuscript that addresses strategies for improving the efficiency of ncAA incorporation by engineering the PKR/Integrated stress response (ISR). There were two remaining concerns. First, inclusion of additional statistical information, which was addressed in the R2. The second prior concern was #4 and the central question- Does the engineered modifications of the ISR have the expected outcomes as viewed by measurements of ISR pathway markers (induced ISR proteins or mRNA, including eIF2alpha-P, total protein synthesis, reporter assays, effector proteins and mRNAs)? The authors indicate that due to complex transfection protocols in the experiments it was not as straightforward as one may presume. New experiments in supplemental figures 16 and 17 included measures of phosphorylated eIF2alpha and total protein synthesis (puromycin incorporation assay). The western blots in Fig. 16A are not fully clear to the eye, but the bar graphs in Fig. 16C suggest that PKR delta thwarts the ISR response, as predicted, and adjustments of eIF2B subunits also lower eIF2alpha phosphorylation for reasons that would not be fully clear to this reviewer. These more direct ISR measurements begin to provide some ISR biomarker support for the ncAA induction of the ISR and remodeling changes as predicted.

Point-by-point

Reviewer comments in black

Our response in blue

Reviewer 2:

This is an R2 manuscript that addresses strategies for improving the efficiency of ncAA incorporation by engineering the PKR/Integrated stress response (ISR). There were two remaining concerns. First, inclusion of additional statistical information, which was addressed in the R2. The second prior concern was #4 and the central question- Does the engineered modifications of the ISR have the expected outcomes as viewed by measurements of ISR pathway markers (induced ISR proteins or mRNA, including eIF2alpha-P, total protein synthesis, reporter assays, effector proteins and mRNAs)? The authors indicate that due to complex transfection protocols in the experiments it was not as straightforward as one may presume. New experiments in supplemental figures 16 and 17 included measures of phosphorylated eIF2alpha and total protein synthesis (puromycin incorporation assay). The western blots in Fig. 16A are not fully clear to the eye, but the bar graphs in Fig. 16C suggest that PKR delta thwarts the ISR response, as predicted, and adjustments of e12B subunits also lower eIF2alpha phosphorylation for reasons that would not be fully clear to this reviewer. These more direct ISR measurements begin to provide some ISR biomarker support for the ncAA induction of the ISR and remodeling changes as predicted.

We thank the reviewer for overall positive comment regarding the experiments we provided in the last submitted version of the manuscript and for the time and effort spent assessing our manuscript.